# Differentially Private Gradient Flow based on the Sliced Wasserstein Distance

**Ilana Sebag**  *i.sebag@criteo.com*
*Criteo AI Lab, Paris, France*
*Miles Team, LAMSADE, Université Paris-Dauphine, PSL University, CNRS, Paris, France*

**Muni Sreenivas Pydi**  *muni.pydi@lamsade.dauphine.fr*
*Miles Team, LAMSADE, Université Paris-Dauphine, PSL University, CNRS, Paris, France*

**Jean-Yves Franceschi**  *jycja.franceschi@criteo.com*
*Criteo AI Lab, Paris, France*

**Alain Rakotomamonjy**  *a.rakotomamonjy@criteo.com*
*Criteo AI Lab, Paris, France*

**Mike Gartrell**  *mike.gartrell@sigmanova.ai*
*Sigma Nova, Paris, France (Work done while working at Criteo AI lab)*

**Jamal Atif**  *jamal.atif@lamsade.dauphine.fr*
*Miles Team, LAMSADE, Université Paris-Dauphine, PSL University, CNRS, Paris, France*

**Alexandre Allauzen**  *alexandre.allauzen@dauphine.psl.eu*
*Miles Team, LAMSADE, Université Paris-Dauphine, PSL University, CNRS, Paris, France*
*ESPCI PSL, Paris, France.*

**Reviewed on OpenReview:** *https://openreview.net/forum?id=aiOHc1LGpD*

## Abstract

Safeguarding privacy in sensitive training data is paramount, particularly in the context of generative modeling. This can be achieved through either differentially private stochastic gradient descent or a differentially private metric for training models or generators. In this paper, we introduce a novel differentially private generative modeling approach based on a gradient flow in the space of probability measures. To this end, we define the gradient flow of the Gaussian-smoothed Sliced Wasserstein Distance, including the associated stochastic differential equation (SDE). By discretizing and defining a numerical scheme for solving this SDE, we demonstrate the link between smoothing and differential privacy based on a Gaussian mechanism, due to a specific form of the SDE's drift term. We then analyze the differential privacy guarantee of our gradient flow, which accounts for both the smoothing and the Wiener process introduced by the SDE itself. Experiments show that our proposed model can generate higher-fidelity data at a low privacy budget compared to a generator-based model, offering a promising alternative.

## 1 Introduction

The widespread use of deep learning in critical applications has heightened concerns about privacy limitations, with various privacy attacks exposing vulnerabilities in machine learning algorithms (Shokri et al., 2017; Hu et al., 2022; Lacharité et al., 2018; Mai et al., 2018), including deep-learning-based generative models (Chen et al., 2020b; Carlini et al., 2023). Differential Privacy (DP) has emerged as a key solution to counter privacy attacks, providing a robust framework to safeguard training data privacy (Dwork et al., 2006; Dwork, 2011;

Dwork & Roth, 2014). DP ensures that a single data point's inclusion or exclusion minimally affects analysis outcomes.

In machine learning, DP is usually achieved by applying calibrated noise to gradient steps involving sensitive training data, with Differentially Private Stochastic Gradient Descent (DP-SGD) being a prominent example (Abadi et al., 2016). While extensively explored in classification, the application of DP in generative models is an emerging research area, often employing DP-SGD variants for training standard generative models (Xie et al., 2018; Chen et al., 2020a; Long et al., 2021; Dockhorn et al., 2023; Ghalebikesabi et al., 2023); either in the context of generator learning (Cao et al., 2021) or diffusion models (Dockhorn et al., 2023).

An under-explored alternative for DP-SGD is to minimize a functional on the space of probability measures:

$$\min_{\mu \in \mathcal{P}(\Omega)} \text{PrivateCost}(\mu, \nu) + \lambda \text{Reg}(\mu), \tag{1}$$

where $\nu$ is the probability measure to be modeled, PrivateCost is a cost functional on the space of probability measures $\mathcal{P}(\Omega)$ that can be computed in a private manner, and Reg is a regularization functional that prevents over-fitting to training samples. Some works solve this problem by using this cost as a loss for training a generator. They employ differentially private versions of metrics such as Maximum Mean Discrepancy or Sliced Wasserstein Distance (Harder et al., 2021; 2023; Rakotomamonjy & Ralaivola, 2021). Here, we propose an alternative line of work that has never been explored, which explicitly minimizes the functional in Eq. (1) using gradient flows.

In a non-DP scenario, gradient flows are commonly employed to address minimization problems like the one described in Eq. (1) (Liutkus et al., 2019; Arbel et al., 2019; Franceschi et al., 2023; Galashov et al., 2024) and are a viable alternative to other generative models (Fan et al., 2022). They possess inherent stability and convergence properties, thereby reaping significant benefits in the optimization process (Ambrosio et al., 2005a;b; Garg & Panagou, 2021). Building on these strengths, gradient flows are particularly well-suited for capturing complex data distributions while maintaining stability during training. For instance, they provide a continuous-time approximation of discrete methods like SGD, facilitating the design of a multi-step push-forward operator that maps a noisy distribution to the target. This approach smooths the optimization path and provides insights into discretization effects. These properties make gradient flows particularly interesting to study in the context of differential privacy. Also, we expect them to enable incorporating differential privacy directly into the metric, rather than relying on DP-SGD as in diffusion models.

In addition, gradient flows have not been previously explored and analyzed within the DP framework. All of the above motivates the following question:

*Can we develop a principled formalism for privacy-preserving generative modeling through gradient flows?*

In this paper we provide an affirmative response to this question by presenting the theoretical framework for a differentially private gradient flow of the sliced Wasserstein distance (SWD).

Our approach involves defining the gradient flow on the smoothed SWD (Rakotomamonjy & Ralaivola, 2021), which is strongly related to DP as made clear latter. Despite its seeming simplicity, Gaussian smoothing, akin to related works on smoothed Wasserstein distance (Goldfeld et al., 2020; Goldfeld & Greenewald, 2020; Ding & Niles-Weed, 2022), introduces theoretical complexities and raises questions about the technical conditions of our gradient flow, including the existence and regularity of its solution. We overcome these challenges and formally establish the continuity equation of the gradient flow of the smoothed SWD, resulting in a smoothed velocity field.

This allows us to choose the discretization of the associated Stochastic Differential Equation (SDE) so that the proposed flow ensures differential privacy. We highlight that after discretization, the smoothing in the drift term acts as a Gaussian mechanism. Furthermore, we show that the discretization further amplifies privacy via the Wiener process in the SDE. This results in a sequential algorithm for which the differential privacy budget can be carefully tracked. We experimentally confirm the viability of our proposed algorithm compared to a baseline generator-based model trained with the same private SWD.

**Notations.** Throughout the paper we use $\Omega$ to denote the sample space. We assume that $\Omega$ is a compact subset of $\mathbb{R}^d$. For any subset $\mathcal{A} \subseteq \mathbb{R}^d$, we use $\mathcal{P}(\mathcal{A})$ to denote the set of probability measures supported on

$\mathcal{A}$ equipped with the Borel $\sigma$-algebra. For $\mu, \nu \in \mathcal{P}(\Omega)$, we use $\Pi(\mu, \nu) \subseteq \mathcal{P}(\Omega^2)$ to denote the set of joint distributions or "couplings" between $\mu$ and $\nu$. For $\mu \in \mathcal{P}(\Omega)$ and a measurable function $M \colon \Omega \to \Omega$, the push-forward operator $\#$ defines a probability measure $M_{\#}\mu \in \mathcal{P}(\Omega)$ such that $M_{\#}\mu(A) = \mu(M^{-1}(A))$ for all measurable $A \subseteq \Omega$. For $r > 0$, we use $\overline{B}(0, r)$ to denote a closed ball of center $0$ and radius $r$.

## 2 Background and Related Work

### 2.1 Sliced Wasserstein Distance

The $p$-Wasserstein distance between two probability measures $\mu, \nu \in \mathcal{P}(\Omega)$ is defined as (Peyré & Cuturi, 2019):

$$\mathcal{W}_p(\mu, \nu) = \left( \inf_{\pi \in \Pi(\mu, \nu)} \int_{\Omega^2} \|x - y\|_2^p \mathrm{d}\pi(x, y) \right)^{\frac{1}{p}}. \tag{2}$$

For $p = 2$, i.e., for the squared Euclidean cost, a celebrated result of Brenier (1991) proves that the optimal way to transport mass from $\mu$ to $\nu$ is through a measure-preserving transport map $M \colon \Omega \to \Omega$ of the form $M(x) = x - \nabla \psi(x)$, where $\psi \colon \Omega \to \Omega$ is a convex function termed the Kantorovich potential between $\mu$ and $\nu$, and $M$ pushes $\mu$ onto $\nu$, i.e. $M_{\#}\mu = \nu$. In the one-dimensional case, the optimal transport map has a closed-form expression given by (Peyré & Cuturi, 2019)

$$M(x) = F_\nu^{-1} \circ F_\mu(x), \tag{3}$$

where $F_\mu$ and $F_\nu$ are the cumulative distribution functions (CDFs) of $\mu$ and $\nu$, respectively. In higher dimensions no such closed form exists.

The sliced Wasserstein distance (SWD) (Rabin et al., 2012) takes advantage of this simplicity of OT in one dimension by computing a distance between $\mu, \nu \in \mathcal{P}(\Omega)$ through their projections $P_{\#}^\theta \mu, P_{\#}^\theta \nu \in \mathcal{P}(\mathbb{R})$ onto the unit sphere $\mathbb{S}^{d-1} = \{\theta \in \mathbb{R}^d \mid \|\theta\|_2 = 1\}$. Here $P^\theta \colon \Omega \to \mathbb{R}$ denotes the projection operator defined as $P^\theta(x) = \langle x, \theta \rangle$. Formally,

$$\mathcal{SW}_2^2(\mu, \nu) = \int_{\mathbb{S}^{d-1}} \mathcal{W}_2^2(P_{\#}^\theta \mu, P_{\#}^\theta \nu) \mathrm{d}\theta, \tag{4}$$

where $\mathrm{d}\theta$ is the uniform probability measure on $\mathbb{S}^{d-1}$. Like $\mathcal{W}_2$, the $\mathcal{SW}_2$ also defines a metric on $\mathcal{P}(\Omega)$ (Nadjahi et al., 2022).

### 2.2 Wasserstein Gradient Flows

A Wasserstein gradient flow represents an extension of the concept of gradient descent applied to a functional within the domain of probability measures. More formally, it constitutes a continuous sequence $(\mu_t)_t$ of probability distributions within a Wasserstein metric space $(\mathcal{P}(\Omega), \mathcal{W}_2)$. $\mathcal{F}_\lambda$ decreases along $(\mu_t)_t$, more formally: the sequence follows a continuity equation (Santambrogio, 2016) with a general form of:

$$\frac{\partial \rho_t}{\partial t} = \mathrm{div}\left(\rho_t \nabla_{W_2} \mathcal{F}_\lambda(\rho_t)\right) = \mathrm{div}\left(\rho_t \nabla_{W_2} \mathcal{F}(\rho_t)\right) + \lambda \Delta \rho_t, \tag{5}$$

where $\mathcal{F}_\lambda = \mathcal{F} + \lambda \mathcal{H}$, $\mathcal{F}$ is the functional to be minimized and $\lambda \mathcal{H}$ is an entropic regularization term. $\mathcal{H}$ is the negative differential entropy ensuring that the model can generalize and avoid overfitting: $\mathcal{H}(\mu) = \int_\Omega \rho(x) \log \rho(x) \mathrm{d}x$. $\lambda \geq 0$ signifies the strength of the entropic regularization. $\rho_t$ is the density of the probability flow $(\mu_t)_{t \geq 0}$ at time $t$. $\nabla_{W_2} \mathcal{F}(\rho_t) \colon \mathbb{R}^d \to \mathbb{R}^d$ is the Wasserstein gradient of $\mathcal{F}$. Depending on the form of $\mathcal{F}_\lambda$, this continuity equation can be associated with a SDE which relates the evolution of $(\mu_t)_{t \geq 0}$ and its particles $(X_t)_{t \geq 0}$ (Jordan et al., 1998):

$$\mathrm{d}X_t = -\nabla_{W_2} \mathcal{F}(\rho_t)(X_t) \mathrm{d}t + \sqrt{2\lambda} \mathrm{d}W_t, \tag{6}$$

where $(W_t)_t$ is a Wiener process, and $X_t \sim \mu_t$.

Wasserstein gradient flows possess a rich theoretical background (Ambrosio, 2008; Santambrogio, 2016). Notably they have been used for generative modeling, where initial distributions are represented by samples that evolve according to a partial differential equation (PDE) governing the gradient flow and defined on several metrics such as the maximum mean discrepancy (Arbel et al., 2019) or Wasserstein-based metrics (Mokrov et al., 2021). Liutkus et al. (2019) also present a non-parametric generative modeling method relying on the gradient flow of the sliced Wasserstein distance, while Bonet et al. (2022) introduce a flow defined in the sliced Wasserstein space by proposing a numerically approximated Jordan–Kinderlehrer–Otto (JKO) type scheme.

In this work we define the gradient flow of the smoothed sliced Wasserstein distance. Smoothing each sliced measure allows us to introduce differential privacy, at the expense of additional theoretical challenges on the definition of the flow.

## 2.3 Differential Privacy

**Definition 1.** *A random mechanism* $\mathcal{M} \colon \mathcal{D} \to \mathcal{R}$ *is* $(\varepsilon, \delta)$*-DP if for any two adjacent inputs* $d_1, d_2 \in \mathcal{D}$ *and any subset of outputs* $\mathcal{S} \subseteq \mathcal{R}$,

$$\mathbb{P}[\mathcal{M}(d_1) \in \mathcal{S}] \leq \mathrm{e}^\varepsilon \mathbb{P}[\mathcal{M}(d_2) \in \mathcal{S}] + \delta. \tag{7}$$

Adjacent inputs refer to datasets differing only by a single record. DP ensures that when a single record in a dataset is swapped, the change in the distribution of model outputs will be controlled by $\varepsilon$ and $\delta$. $\varepsilon$ controls the trade-off between the level of privacy and the usefulness of the output, where smaller $\varepsilon$ values offer stronger privacy but potentially lower utility (e.g. in our specific case, low-quality generated samples). $\delta$ is a bound on the external risk (e.g. information is accidentally being leaked) that won't be restricted by $\varepsilon$; it is an extra privacy option that enables control of the extent of the privacy being compromised. In practice, we are interested in the values of $\delta$ that are less than the inverse of a polynomial in the size of the database (Dwork & Roth, 2014).

A classical example of a DP mechanism is the Gaussian mechanism operating on a function $f \colon \mathcal{D} \to \mathbb{R}^d$ as:

$$\mathcal{M}_f(d) = \mathcal{N}(f(d), \sigma^2 \mathcal{I}_d). \tag{8}$$

We define the $\ell_2$ sensitivity of $f$ as $\Delta_2(f) := \max_{d_1, d_2 \colon \text{adjacent} \in \mathcal{D}} \|f(d_1) - f(d_2)\|_2$. For $c^2 > 2\ln(1.25/\delta)$ and $\sigma \geq c \frac{\Delta_2(f)}{\varepsilon}$, the Gaussian mechanism is $(\varepsilon, \delta)$-DP (Dwork & Roth, 2014).

Several works deal with differentially private generative modeling, with most adopting DP-SGD (Xie et al., 2018; Chen et al., 2020a; Long et al., 2021; Dockhorn et al., 2023; Ghalebikesabi et al., 2023). This approach is commonly employed in the context of generator learning (Cao et al., 2021) and diffusion models (Dockhorn et al., 2023).

An alternative solution for a DP generative model is to consider a DP loss function on which the generator is trained. However, there is limited research on defining rigorous metrics that can be computed in a private manner, resulting in a gap for privacy-preserving machine learning algorithms. While Lê Tien et al. (2019) use random projections to make $\mathcal{W}_1$ computation differentially private, they do not provide a theoretical analysis of the resulting divergence. Instead, Harder et al. (2021; 2023) have considered differentially private Maximum Mean Discrepancy as a generator loss. Rakotomamonjy & Ralaivola (2021) introduce a Gaussian-smoothed version of $\mathcal{SW}_2$, defined as

$$\mathcal{G}_\sigma \mathcal{SW}_2^2(\mu, \nu) = \int_{\mathbb{S}^{d-1}} \mathcal{W}_2^2(P_\#^\theta \mu * \xi_\sigma, P_\#^\theta \nu * \xi_\sigma) \mathrm{d}\theta, \tag{9}$$

where $\xi_\sigma \sim \mathcal{N}(0, \sigma^2 \mathcal{I}_d)$. They demonstrate that this distance and some extensions (Rakotomamonjy et al., 2021) are inherently differentially private, as the smoothing acts as a Gaussian mechanism. This allows for the seamless integration of this differentially private loss function into machine learning problems involving distribution comparisons, such as generator-based generative modeling. In this work, we extend this trend by formalizing the gradient flow of the Gaussian-smoothed Sliced Wasserstein Distance.

# 3 Differentially Private Gradient Flow

In this section we present the theoretical building blocks of our method, which consists of building a discretized gradient flow on the smoothed sliced Wasserstein distance. This smoothing and discretization will lead to a differentially private drift (vector field) in the gradient flow. In Section 3.1 we introduce the smoothing and the gradient flow of the Gaussian-smoothed sliced Wasserstein distance of Rakotomamonjy & Ralaivola (2021) defined in Eq. (9), and prove the existence and regularity of its solution. In Section 3.2 we present a particle scheme to simulate the discretization of this flow and elaborate the link between the discretization, smoothing, and differential privacy. In Section 3.3 we present the privacy guarantee.

## 3.1 Gradient Flow of the Smoothed Sliced Wasserstein Distance

In this subsection we study the following functional over the Wasserstein space $(\mathcal{P}(\Omega), \mathcal{W}_2)$:

$$\mathcal{F}^{\nu}_{\lambda,\sigma}(\mu) = \frac{1}{2}\mathcal{G}_\sigma \mathcal{SW}_2^2(\mu, \nu) + \lambda \mathcal{H}(\mu), \tag{10}$$

where $\nu \in \mathcal{P}(\Omega)$ is the target distribution to be modeled and $\sigma > 0$ is the smoothing of the probability measures in the inner optimal transport problem in $\mathcal{G}_\sigma \mathcal{SW}_2$.

We will show later how $\lambda$ and $\sigma$ relate to the privacy parameters $(\varepsilon, \delta)$ in the discretized flow. The main result of this subsection is the following: first, we establish the existence and regularity of a Generalized Minimizing Movement Scheme (GMMS) for Eq. (10), and then we demonstrate that this GMMS satisfies the corresponding continuity equation.

**Theorem 1.** *Let $\nu \in \mathcal{P}(\overline{B}(0, r))$ have a strictly positive smooth density. For $\lambda > 0$ and $r > \sqrt{d}$, let the starting distribution $\mu_0 \in \mathcal{P}(\overline{B}(0, r))$ have a density $\rho_0 \in \mathrm{L}^\infty(\overline{B}(0, 1))$. There exists a minimizing movement scheme $(\mu_t)_{t \geq 0}$ associated with Eq. (10). Further, $(\mu_t)_{t \geq 0}$ admits densities $(\rho_t)_{t \geq 0}$ following a continuity equation:*

$$\frac{\partial \rho_t}{\partial t} = -\mathrm{div}(v_t^{(\sigma)} \rho_t) + \lambda \Delta \rho_t, \tag{11}$$

*with:*

$$v_t^{(\sigma)}(x) = v^{(\sigma)}(x, \mu_t) = \int_{\mathbb{S}^{d-1}} (\psi_{\mu_t,\theta}^{(\sigma)})'(\langle x, \theta \rangle) \theta \mathrm{d}\theta. \tag{12}$$

*Here, $\psi_{\mu_t,\theta}^{(\sigma)}$ is the Kantorovich potential (see Section 2.1) between $P_\#^\theta \mu_t * \xi_\sigma$ and $P_\#^\theta \nu * \xi_\sigma$, with $\xi_\sigma \sim \mathcal{N}(0, \sigma^2)$, and its derivative is given by Brenier (1991):*

$$(\psi_{\mu_t,\theta}^{(\sigma)})'(z) = z - F_{P_\#^\theta \mu_t * \xi_\sigma}^{-1} \circ F_{P_\#^\theta \nu * \xi_\sigma}(z), \tag{13}$$

*where $F_\rho$ denotes the CDF of $\rho$.*

*Proof sketch.* **(1)** We begin by showing that there exists a GMMS (see Appendix A for the precise definition) for the functional in Eq. (11). For this we show that the following optimization problem admits a solution for any $h > 0$:

$$\mu_{k+1}^h \in \operatorname*{arg\,min}_{\mu \in \mathcal{P}(\Omega)} \mathcal{F}^{\nu}_{\lambda,\sigma}(\mu) + \frac{\mathcal{W}_2^2(\mu, \mu_k^h)}{2h}. \tag{14}$$

Notice that the above problem is simply the implicit Euler scheme for deriving the gradient flow of $\mathcal{F}^{\nu}_{\lambda,\sigma}$ over the Wasserstein space $(\mathcal{P}(\Omega), \mathcal{W}_2)$. Since $\mathcal{P}(\overline{B}(0, 1))$ is compact for weak convergence, it is enough to show that $\mathcal{F}^{\nu}_{\lambda,\sigma}$ is lower semi-continuous in $\mathcal{W}_2$. By Lemma 9.4.3 of Ambrosio (2008), $\mathcal{H}$ is lower semi-continuous. By Rakotomamonjy & Ralaivola (2021), $\mathcal{G}_\sigma \mathcal{SW}_2(\mu, \nu)$ is symmetric and satisfies the triangle inequality. Moreover, $\mathcal{G}_\sigma \mathcal{SW}_2(\mu, \nu) \leq \mathcal{SW}_2(\mu, \nu)$ for any $\sigma \geq 0$. Hence for any $\xi, \xi' \in \mathcal{P}(\overline{B}(0, 1))$,

$$|\mathcal{G}_\sigma \mathcal{SW}_2(\xi, \nu) - \mathcal{G}_\sigma \mathcal{SW}_2(\xi', \nu)| \leq \mathcal{G}_\sigma \mathcal{SW}_2(\xi, \xi') \leq \mathcal{SW}_2(\xi, \xi') \leq c_d \mathcal{W}(\xi, \xi'), \tag{15}$$

where $c_d > 0$ is a constant only dependent on the dimension $d$ and the last inequality follows from Prop. 5.1.3 in Bonnotte (2013). Hence there exists a minimum $\hat{\mu} \in \mathcal{P}(\overline{B}(0, r))$ of $\mathcal{G}(\mu)$, admitting a density $\hat{\rho}$, because otherwise $\mathcal{H}(\hat{\mu}) = \infty$. Further, we prove that the solution is "sufficiently regular" in Lemma 3 of Appendix A. Due to the distinct nature of the $\mathcal{G}_\sigma \mathcal{SW}$ metric, proving the existence and regularity of the solution has never been done before and constitutes a novel contribution. To establish regularity (Lemma 3), we adopt proof strategies inspired by Bonnotte (2013)'s previous work, but we need to adjust them to fit our specific $\mathcal{G}_\sigma \mathcal{SW}$ case, requiring a complete construction from scratch.

**(2)** Next, we show that the GMMS whose existence and regularity were previously established indeed satisfies the continuity equation in Eq. (11). A crucial ingredient in this step is the analysis of the first variation of the Gaussian-smoothed $\mathcal{SW}_2$ distance which is given in the following proposition (see Appendix A for proof).

**Proposition 1.** *Let $\mu, \nu \in \mathcal{P}(\Omega)$. For any diffeomorphism $\zeta$ of $\Omega$,*

$$\lim_{\varepsilon \to 0^+} \frac{\mathcal{G}_\sigma \mathcal{SW}_2^2([\mathrm{Id} + \varepsilon \zeta]_\sharp \mu, \nu) - \mathcal{G}_\sigma \mathcal{SW}_2^2(\mu, \nu)}{2\varepsilon} = \fint_{\mathbb{S}^{d-1}} \int_\Omega (\psi_{\mu_t,\theta}^\sigma)'(\langle \theta, x \rangle) \langle \theta, \zeta(x) \rangle \mathrm{d}\mu \mathrm{d}\theta. \tag{16}$$

Using the above result we then get the desired flow equation by closely following the proofs of Theorem S6 in Liutkus et al. (2019) and Theorem 5.6.1 in Bonnotte (2013). □

Theorem 1 shows that there is a continuous sequence of probability measures $(\mu_t)_{t \geq 0}$ that constitutes a minimizing movement scheme for the functional in Eq. (10). Moreover, it shows that the probability density $\rho_t$ of the minimizing movement scheme satisfies the continuity equation given by Eq. (11). Theorem 1 is a generalization of Theorem 2 of Liutkus et al. (2019) and Theorem 5.6.1 of Bonnotte (2013), which we retrieve when $\sigma \to 0$. However, the proof does not trivially follow as a corollary from these results, since we consider projected and *then* smoothed measures instead of directly applying previous results on smoothed measures before projection. Hence, we need two new pieces in the proof that are not present in prior works: **(1)** existence and regularity of solutions to the functional minimization problem, and **(2)** analysis of the first variation of the squared Gaussian-smoothed SW metric.

One key point of our approach is that the drift term in the continuity equation relies on the Kantorovich potential $\psi_{\mu_t,\theta}^{(\sigma)}$ associated with the convolution of the Gaussian-smoothed and projected measures, specifically $P_\#^\theta \mu_t * \xi_\sigma$ and $P_\#^\theta \nu * \xi_\sigma$. When transitioning from the continuous PDE to a discrete-time SDE this Gaussian smoothing becomes crucial, ultimately leading to the Gaussian mechanism and ensuring differential privacy. We show in the next subsection that the combination of smoothing in projected measures and the discretization jointly establishes differential privacy.

## 3.2 Discretized and Private Gradient Flow Algorithms

In this subsection, we show how the gradient flow of Theorem 1 is discretized and how the smoothing acts as a Gaussian mechanism. We then present two algorithms that implement this gradient flow.

The gradient flow of $\mathcal{G}_\sigma \mathcal{SW}_2$ given in Eq. (11) corresponds to a nonlinear Fokker-Plank type equation, where the drift term is dependent on the density of the solution. The evolution of $(\mu_t)_{t \geq 0}$ in Eq. (11) corresponds to a stochastic process $(X_t)_{t \geq 0}$ that solves the SDE in Eq. (6). The latter can be discretized using the Euler-Maruyama scheme with the random variable initialization $\widehat{X}_0 \sim \mu_0$ as follows:

$$\widehat{X}_{k+1} = h v^{(\sigma)}(\widehat{X}_k, \widehat{\mu}_{kh}) + \sqrt{2\lambda h} Z_{k+1}, \tag{17}$$

where $h > 0$ is the step size, $\widehat{\mu}_{kh}$ is the distribution of $\widehat{X}_k$, and $\{Z_k\}_k$ are i.i.d. standard normal random variables. The discrete-time SDE in Eq. (17) can then be simulated by a stochastic finite particle system $\{\widehat{X}_k^i\}$, where $i \in \{1, \dots, N\}$ is the index for the $i^{\mathrm{th}}$ particle, and the discrete time index $k$ runs from 0 to $Kh = T$ (Bossy & Talay, 1997). The particles are initialized as $\widehat{X}_0^i \sim \mu_0$ i.i.d., where each particle follows the update equation:

$$\widehat{X}_{k+1}^i = h \widehat{v}^{(\sigma)}(\widehat{X}_k^i, \hat{\mu}_{kh}^i) + \sqrt{2\lambda h} Z_{k+1}^i, \tag{18}$$

with $\widehat{v}^{(\sigma)}(\widehat{X}_k^i, \widehat{\mu}_{kh}^i)$ being an estimate of $v^{(\sigma)}(\widehat{X}_k, \widehat{\mu}_{kh})$; cf. Eq. (20) for the details.

Naturally, approximating the continuous time SDE in Eq. (6) by the discrete-time update in Eq. (18) leads to some error, which we provide bounds for in the following theorem. In the infinite particle regime, i.e., as $N \to \infty$, under some assumptions of regularity and smoothness of the drift terms $v^{(\sigma)}$ and $\widehat{v}^{(\sigma)}$, we can state the following.

**Theorem 2.** *Suppose that the SDE in Eq. (6) has a unique strong solution[1] $(X_t)_{t \in [0,T]}$ for any starting point $x_0 \in \Omega$, such that $X_T \sim \mu_T$. For $T = Kh$, let $\widehat{\mu}_{Kh}$ be the distribution of $\widehat{X}_K$ in the discrete-time SDE in Eq. (17). Under suitable assumptions of regularity and Lipschitzness on $v^{(\sigma)}$ and $\widehat{v}^{(\sigma)}$ stated in the Appendices, the following bound holds for any $\lambda > TL^2/8$:*

$$\|\mu_T - \widehat{\mu}_{Kh}\|_{TV}^2 \le \frac{T}{\lambda - TL^2/8} \left[ L^2 h(c_1 h + d\lambda) + c_2 \kappa \right], \tag{19}$$

*where $c_1, c_2, L, \kappa > 0$ are constants independent of time (but may depend on $\sigma$).*

The above theorem follows in a straightforward manner from Theorem 3 of Liutkus et al. (2019) for the $\mathcal{SW}_2$ flow. It is worth noting that the error bound in Eq. (19) is possibly tighter than the corresponding error bound in Theorem 3 of Liutkus et al. (2019) for the special case of $\sigma \to 0$, because the constant $L$ can be shown to be a non-increasing function of $\sigma$. The error bound depends linearly on the dimension $d$ and will tend to 0 for $\kappa = 0$ and small step size $h$.

We now delve into the numerical computation of the particle flow in Eq. (18). To evaluate $\widehat{v}^{(\sigma)}$, we need two approximations. The first one approximates the distribution $\widehat{\mu}_{kh}$ by the empirical distribution of the particles at time $k$, $\widehat{\mu}_{kh} \approx \widehat{\mu}_{kh}^{(N)} := \frac{1}{N} \sum_{i=1}^N \delta_{\widehat{X}_k^i}$ where here ($\delta$ is a Dirac distribution). The second one replaces the integral over $\theta \in \mathbb{S}^{d-1}$ in Eq. (12), with a Monte Carlo estimate, for $\theta_j$ a set of projections drawn from the sphere $\mathbb{S}^{d-1}$:

$$\widehat{v}_k^{(\sigma)}(x) := -\frac{1}{N_\theta} \sum_{j=1}^{N_\theta} (\psi_{\mu_k, \theta_j}^{(\sigma)})'(\langle x, \theta_j \rangle) \theta_j, \tag{20}$$

where $(\psi_{\mu_k, \theta_j}^{(\sigma)})'$ is the discretized derivative of the Kantorovich potential between $P_\#^{\theta_j} \widehat{\mu}_{kh}^{(N)} * \xi_\sigma$ and $P_\#^{\theta_j} \nu * \xi_\sigma$, with $\xi_\sigma \sim \mathcal{N}(0, \sigma^2)$, and $(\psi_{\mu_k, \theta_j}^{(\sigma)})'(z)$ is defined as (Brenier, 1991):

$$(\psi_{\mu_k, \theta_j}^{(\sigma)})'(z) = z - F_{P_\#^{\theta_j} \widehat{\mu}_{kh}^{(N)} * \xi_\sigma}^{-1} \circ F_{P_\#^{\theta_j} \nu * \xi_\sigma}(z) \tag{21}$$

This equation is key in the gradient flow since it is the main block of the drift term in Eq. (20) and it plays an essential role in bridging the smoothing and the privacy. Indeed, since convolution of probability distributions boils down to the addition of random variables, Eq. (21) can be considered as a Gaussian mechanism (see Section 2.3) applied to the projected measures. In practice, we compute $P_\#^{\theta_j} \widehat{\mu}_{kh}^{(N)} * \xi_\sigma$ and $P_\#^{\theta_j} \nu * \xi_\sigma$ in the following way. Let $\Theta = [\theta_1^T, \ldots, \theta_{N_\theta}^T] \in \mathbb{R}^{d \times N_\theta}$ be the *random projection matrix* composed of all the $N_\theta$ projection vectors sampled uniformly from $\mathbb{S}^{d-1}$ and $X = [x_1^T, \ldots, x_n^T]^T, Y = [y_1^T, \ldots, y_n^T]^T \in \mathbb{R}^{n \times d}$, respectively, the *data matrices* composed of the $n$ i.i.d. samples from the target distribution $\nu$ and the empirical distribution $\widehat{\mu}_{kh}^{(N)}$. Then, the smoothed and projected distributions $P_\#^{\theta_j} \nu * \xi_\sigma$ and $P_\#^{\theta_j} \widehat{\mu}_{kh}^{(N)} * \xi_\sigma$ can be respectively written as $X\Theta + Z_X$ and $Y\Theta + Z_Y$ with $Z_X, Z_Y \in \mathbb{R}^{n \times N_\theta}$ being the i.i.d. Gaussian random variables with variance $\sigma^2$, corresponding to the Gaussian mechanism.

From an algorithmic point of view, the random projection matrix $\Theta \in \mathbb{R}^{d \times N_\theta}$ can either be sampled at the start of every discrete time step, or sampled once at the beginning of the algorithm and reused in every

---

[1]A strong solution to an SDE means that it is defined over a given probability space and Wiener process, while a weak solution would be defined over some probability space and Wiener process. The interested reader can refer to e.g. (Øksendal, 2000).

---

**Algorithm 1:** DP Sliced Wasserstein Flow with resampling of $\theta$'s: DPSWflow-r.

---

1: **Input:** $Y = [y_1^T, \ldots, y_n^T]^T \in \mathbb{R}^{n \times d}$ i.e. $N$ i.i.d. samples from target distribution $\nu$, number of projections $N_\theta$, regularization parameter $\lambda$, variance $\sigma$, step size $h$.

2: **Output:** $X = [x_1^T, \ldots, x_n^T]^T \in \mathbb{R}^{n \times d}$

3: *// Initialize the particles*

4: $\{x_i\}_{i=1}^N \sim \mu_0$, $X = [x_1^T, \ldots, x_n^T]^T \in \mathbb{R}^{n \times d}$

5: *// Iterations of the flow*

6: **for** $k = 0, \ldots, K-1$ **do**

7: $\quad \{\theta_j\}_{j=1}^{N_\theta} \sim \mathrm{Unif}(\mathbb{S}^{d-1})$, $\Theta = [\theta_1^T, \ldots, \theta_{N_\theta}^T]$

8: $\quad$ Sample $Z_X, Z_Y \in \mathbb{R}^{n \times N_\theta}$ from i.i.d. $\mathcal{N}(0, \sigma^2)$

9: $\quad$ Compute the inverse CDF of $Y\Theta + Z_Y$

10: $\quad$ Compute the CDF of $X\Theta + Z_X$

11: $\quad$ Compute $\widehat{v}_k^{(\sigma)}(x_i)$ using Eq. (20)

12: $\quad x_i \leftarrow h\widehat{v}_k^{(\sigma)}(x_i) + \sqrt{2\lambda h}z$, $z \sim \mathcal{N}(0, \mathcal{I}_d)$

13: **end for**

---

step. These two strategies lead to DPSWflow-r (Algorithm 1) and DPSWflow (Algorithm 2) (the latter is detailed in Appendix C.3). The main difference between both lies in the sampling of the projections $\theta$ of the sliced Wasserstein. In DPSWflow-r, we resample $N_\theta$ projections at each iteration of the flow leading to a more expensive iteration. In DPSWflow, we sample all the $N_\theta$ projections in advance (typically a larger amount) and use them in all subsequent iterations by subsampling; *e.g* at each iteration, we subsample a set of projections among the pre-computed $N_\theta$ projections. This enables us to save on the privacy budget as it implies "free iterations" in term of privacy. The choice between resampling or pre-sampling projections was not explicit in the prior non-DP work of Liutkus et al. (2019). They adopt pre-sampling in the provided algorithm of their paper, whereas they resample projections in their code. In contrast, in our paper, we explicitly highlight the importance of this choice, for both privacy and performance issues.

Before clarifying this difference, we provide details regarding how privacy guarantees are dealt with in the gradient flow.

### 3.3 Privacy Guarantee

In this subsection we analyze the DP guarantee of the particle scheme outlined in the previous subsection. In our gradient flow algorithm, there exist two independent levels of Gaussian mechanisms: (i) in the drift term, at the level of random projection of data $X$ through the matrix $\Theta$ *and* (ii) at the addition of the diffusion term $\sqrt{2\lambda h}Z$ in the particle update Eq. (18). Then, we comment on how privacy impacts each of our particle flows in Algorithms 1 and 2, defined by Eq. (18),

#### 3.3.1 Privacy guarantee arising from the random projection

To track the privacy guarantee arising from the random projection, we define a randomized mechanism $\mathcal{M}_{N_\theta, \sigma} : \mathbb{R}^{n \times d} \to \mathbb{R}^{n \times N_\theta}$ as:

$$\mathcal{M}_{N_\theta, \sigma}(X) = X\Theta + Z_\sigma, \tag{22}$$

where $\Theta$ is the random projection matrix and $Z_\sigma \in \mathbb{R}^{n \times N_\theta}$ consists of i.i.d. Gaussian random variables with variance $\sigma^2$. Given $X$ composed of $\{x_i\}_i \sim \nu$, the position of these particles can be updated based on the drift of Eq. (20) which leverages $\mathcal{M}_{N_\theta, \sigma}(X)$ for computing Eq. (21). Hence, if $\mathcal{M}_{N_\theta, \sigma}(X)$ is $(\varepsilon, \delta)$-DP, then by the post-processing property of DP (Dwork et al., 2006), the computation of the drift term for all $n$ particles in one time step is also $(\varepsilon, \delta)$-DP. To derive the DP guarantee for $\mathcal{M}_{N_\theta, \sigma}(X)$, we use the following lemma.

**Lemma 1** (Rakotomamonjy & Ralaivola (2021)). *For data matrices $X, X' \in \mathbb{R}^{n \times d}$ that differ in only the $i^{th}$ row, satisfying $\|X_i - X_i'\|_2 \le 1$, and a random projection matrix $\Theta \in \mathbb{R}^{d \times N_\theta}$ whose columns are randomly sampled from $\mathbb{S}^{d-1}$, and $N_\theta$ being large enough ($N_\theta > 30$) , the following bound holds with probability at least*

$1 - \delta$:

$$\|X\Theta - X'\Theta\|_F^2 \leq w(N_\theta, \delta), \quad with, \quad w(N_\theta, \delta) = \frac{N_\theta}{d} + \frac{z_{i-\delta}}{d}\sqrt{\frac{2N_\theta(d-1)}{d+2}}, \tag{23}$$

*where $z_{i-\delta} = \Phi^{-1}(1-\delta)$ and $\Phi$ is the CDF of a zero-mean unit variance Gaussian distribution.*

### 3.3.2 Privacy guarantee arising from the diffusion term

To track the privacy guarantee arising from the diffusion term, we define a Markov operator $\mathcal{K}_{h,\lambda}\colon \mathbb{R}^{n \times d} \to \mathbb{R}^{n \times d}$ as

$$\mathcal{K}_{h,\lambda}(X) = \left[h\widehat{v}_k^{(\sigma)}(X^i) + \sqrt{2\lambda h}Z\right]_{i=1}^n, \tag{24}$$

where $\widehat{v}_k^{(\sigma)}(x)$ is the drift term as defined in Eq. (20) and $Z \sim \mathcal{N}(0, I_d)$. The following lemma from Balle et al. (2019) can then be used to characterize the privacy guarantee resulting from both sources of randomness.

**Lemma 2** (Balle et al. (2019)). *Let $\mathcal{K}\colon X \to Y$ be a Markov operator satisfying the following condition for any $x, x' \in X$:*

$$\|\mathcal{K}(x) - \mathcal{K}(x')\|_{\mathrm{TV}} \leq \gamma. \tag{25}$$

*Then for any $(\varepsilon, \delta)$-DP randomized mechanism $\mathcal{M}$, $\mathcal{K} \circ \mathcal{M}$ is $(\varepsilon, \gamma\delta)$-DP.*

Leveraging Lemmas 1 and 2, we get the following theorem on the privacy guarantee of Algorithms 1 and 2.

**Theorem 3.** *Under the setup of Lemma 1, the particle update in Eq. (18) is $\left(\frac{cw(N_\theta, \delta)}{\sigma}, \sqrt{\frac{h}{2\lambda}}\delta\right)$-DP, where $c$ is a constant satisfying $c^2 > 2\ln(1.25/\delta)$.*

*Proof.* For data matrices $X, X' \in \mathbb{R}^{n \times d}$ that differ in only the $i^{\text{th}}$ row, satisfying $\|X_i - X'_i\|_2 \leq 1$,

$$D_{\mathrm{KL}}\left(\mathcal{K}_{h,\lambda}(X), \mathcal{K}_{h,\lambda}(X')\right) = D_{\mathrm{KL}}\left(\mathcal{N}(h\widehat{v}^{(\sigma)}(X_i), 2h\lambda I_d), \mathcal{N}(h\widehat{v}^{(\sigma)}(X'_i), 2h\lambda I_d)\right)$$

$$= \frac{1}{2(2h\lambda)}\|h\widehat{v}^{(\sigma)}(X_i) - h\widehat{v}^{(\sigma)}(X'_i)\|^2 \leq \frac{h}{\lambda}, \tag{26}$$

where the last inequality follows from the observation that $\|\widehat{v}^{(\sigma)}(X_i))\| \leq 1$. Hence,

$$\|\mathcal{K}_{h,\lambda}(X) - \mathcal{K}_{h,\lambda}(X')\|_{TV} \leq \sqrt{\frac{1}{2}D_{\mathrm{KL}}\left(\mathcal{K}_{h,\lambda}(X), \mathcal{K}_{h,\lambda}(X')\right)} \leq \sqrt{\frac{h}{2\lambda}}. \tag{27}$$

Plugging in the sensitivity bound from Lemma 1 into the Gaussian mechanism presented in Section 2.3, we see that the mechanism $\mathcal{M}_{N_\theta, \sigma}$ is $(\varepsilon, \delta)$-DP for $\sigma = \frac{cw(N_\theta, \delta)}{\varepsilon}$. The desired result then follows from an application of Lemma 2 to the post-processed mechanism $\mathcal{K}_{h,\lambda} \circ \mathcal{M}_{N_\theta, \sigma}$. □

We observe that the privacy parameter $\varepsilon$ degrades linearly with the number of projections $N_\theta$, while the $\delta$ parameter decreases with the step size of the discretization. The sensitivity depends linearly on $\frac{1}{\sqrt{d}}$. Thus, the higher the $d$, the more private the mechanism is.

### 3.3.3 On the impact of (re-)sampling on privacy

Algorithms 1 and 2 differ by one element: in the former we choose a small $N_\theta$ which is being resampled at each step of the flow, whilst in the latter we choose a large $N_\theta$ among which we sub-sample during the flow. The idea of reusing the random projections between iterations in Algorithm 1 has an important effect on the resulting performance at similar privacy guarantees. Our intuition is that the DP bound is tighter for DPSWflow-r as we are allowed to use the moment accountant (Abadi et al., 2016) to derive the DP bound: this composition of DP operations is cheaper in privacy than doing them all at once at the beginning of the gradient flow. More details on the privacy guarantees of Algorithm 1 and Algorithm 2 are given in Appendix C.4. Also, resampling enables unbiased gradient estimates across iterations, which, we suppose, is important for generation performance. Experimental results are reported for both mechanisms in Section 4.5.

# 4 Experiments

In this section we evaluate our method within a generative modeling context. The primary objective is to validate our theoretical framework and showcase the behavior of our approach, rather than strive for state-of-the-art results in generative modeling. Our study focuses on a specific claim: in a privacy setting, a gradient flow performs better than generator-based models trained with the same metric. The code for the experiments can be found at https://github.com/ilanasebag/dpswgf.

## 4.1 Toy Problem

We use a toy problem to illustrate how our differentially private sliced Wasserstein gradient flow behaves compared to the vanilla non-DP baseline. Here we set $N_\theta = 200$, $h = 1$, and $\lambda = 0.001$. Examples are provided in Fig. 1. We see that the particle flow of the sliced Wasserstein can correctly approximate the target distributions that are composed of 5 Gaussians, as measured by SWD. With the Gaussian smoothing, for $\sigma = 0.5$, the particles are still able to match the target distribution, although the samples are more dispersed than for the noiseless SWF, leading to a SWD value of 0.81 instead of 0.08. Finally, for $\sigma = 1$, our approach struggles in matching the true distribution, although many particles are still within its level sets.

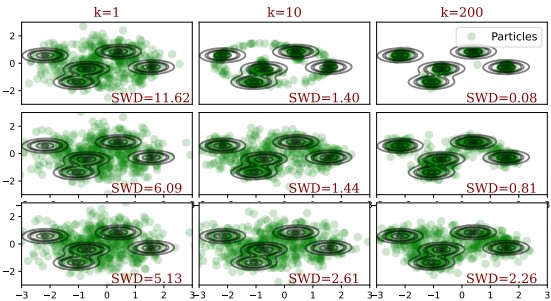

Figure 1: Examples of particle flows for (top) sliced Wasserstein flow, (middle) our DPSWflow with $\sigma = 0.5$, and (bottom) $\sigma = 1$. Each panel shows the level sets (in black) of the target distribution, which is composed of 5 Gaussians, as well as the particles (in green). The columns depict the particles after the (left) first step, (middle) 10-th, and (right) the 200-th steps of the flow.

## 4.2 Comparisons and Baselines

To demonstrate our claims we test our method on three mechanisms: DPSWflow-r (Algorithm 1), DPSWflow (Algorithm 2) and DPSWgen. DPSWgen is a generator-based model adapted from Rakotomamonjy & Ralaivola (2021) trained using the differentially private sliced Wasserstein distance. While both DPSWgen and our method use the same metric to measure distributional similarity, they differ in how samples are generated: DPSWgen employs a one-step generative model that maps Gaussian noise directly to the target distribution. In contrast, our method generates samples through a gradient flow, iteratively transitioning from a Gaussian distribution to the target distribution, acting as a multi-step generator. Therefore, it is differentially private because its drift term is built through Gaussian mechanism. Both approaches are fundamentally distinct.

To maintain a suitably low input space dimension, in order to mitigate the curse of dimensionality and reduce computational cost, our mechanisms 1 and 2 are preceded by an autoencoder with $\mathcal{Z}_\mu$ as the latent space. Subsequently, they take the latent space $\mathcal{Z}_\mu \subseteq \mathbb{R}^d$ of the autoencoder as the input space, and ensure differential privacy using the DP gradient flow. To ensure a fair comparison, DPSWgen is also preceded by the same autoencoder with $\mathcal{Z}_\mu$ as the latent space. Subsequently, it also takes the latent space $\mathcal{Z}_\mu \subseteq \mathbb{R}^d$ of the autoencoder as the input space.

We evaluate the three algorithms using the Fréchet inception distance (FID, Heusel et al., 2018). In our results, we present each method at three levels of differential privacy: $\varepsilon = \infty$ (no privacy), $\varepsilon = 10$, and $\varepsilon = 5$,

along with their corresponding FID scores. In this context the optimal generated images from each model are expected to yield the best achievable FID score when $\varepsilon = \infty$.

### 4.3 Sensitivity and Privacy Budget Tracking

For both DPSWflow-r and DPSWgen we monitor the privacy budget using the Gaussian moments accountant method proposed by Abadi et al. (2016), where we choose a range of $\sigma$'s satisfying the constraint $\sigma \geq \frac{cw(N_\theta, \delta)}{\varepsilon}$, where $w$ is the sensitivity bound from Lemma 1, $c > 2\ln(1.25/\delta)$, and we use the moment accountant to obtain the corresponding $\varepsilon$'s. Also, to prevent privacy leakage, we normalize the latent space of the autoencoder (which is used as input of the flow and the generator) to norm 1, so we incur an additional factor of 2 in the sensitivity bound.

### 4.4 Settings and Datasets

In all three DPSWflow, DPSWflow-r, and DPSWgen models, we pre-train an autoencoder and then use a DP sliced Wasserstein flow component, or DP generator, respectively. In order to uphold the integrity of the differential privacy framework and mitigate potential privacy breaches, we conducted separate pre-training procedures for the autoencoder and the flows / generator using distinct datasets: a publicly available dataset for the autoencoder, and a confidential dataset for the flows / generator. In practice, we partitioned the training set $X$ into two distinct segments of equal size, denoted as $X^{\mathrm{pub}}$ and $X^{\mathrm{priv}}$. Subsequently, we conducted training of the autoencoder on $X^{\mathrm{pub}}$. Then, we compute the encoded representation on $X^{\mathrm{priv}}$ in the latent space and use it as input to DPSWflow, DPSWflow-r, and DPSWgen. Furthermore, as mentioned in Section 4.3, to prevent privacy leakage we normalize the latent space before adding Gaussian noise, ensuring that the encoded representations lie on a hypersphere.

We assessed each method on three datasets: MNIST (LeCun et al., 1998), FashionMNIST (F-MNIST, Xiao et al., 2017), and CelebA (Liu et al., 2015). The experiments performed on the MNIST and FashionMNIST datasets use the same autoencoder architecture as the framework proposed by Liutkus et al. (2019). The experiments conducted on the CelebA dataset use an autoencoder architecture adapted from a DCGAN (Radford et al., 2016b). More details on these architectures are given in Appendix C.2.

### 4.5 Experimental Results

This section outlines our experimental findings, including the resulting FID scores (Table 1) and the generated samples for each of our experiments (Figures Figs. 2 to 4). We compute the FID as the average of the FID scores obtained from five generation runs. Also, for each experiment, we use $\delta > 0$: $\delta = 10^{-5}$ for MNIST and FashionMNIST, and $\delta = 10^{-6}$ for CelebA. Fig. 2 presents the results from our model and the baseline without any differential privacy applied. Figs. 3 and 4 show the results for $\varepsilon = 10$ and $\varepsilon = 5$, respectively.

Fig. 2 presents the results from our model and the baseline without any differential privacy applied. Figs. 3 and 4 show the results for $\varepsilon = 10$ and $\varepsilon = 5$, respectively.

We observe that our methods outperform the DPSWgen baseline for all privacy budgets tested in our experiments, both in terms of FID scores and the visual quality of the generated samples. Furthermore, the variant of our approach with resampling (DPSWflow-r) consistently outperforms the variant without

Table 1: FID results for each baseline, dataset and privacy setting, averaged over 5 generation runs.

|  | MNIST | | | F-MNIST | | | CELEBA | | |
|---|---|---|---|---|---|---|---|---|---|
| $\varepsilon$ | $\infty$ | 10 | 5 | $\infty$ | 10 | 5 | $\infty$ | 10 | 5 |
| DPSWgen | 114 | 124 | 198 | 138 | 170 | 199 | 171 | 209 | 214 |
| DPSWflow-r | 21 | 70 | 114 | 42 | 88 | 98 | 57 | 132 | 197 |
| DPSWflow | 73 | 118 | 171 | 96 | 98 | 129 | 134 | 262 | 292 |

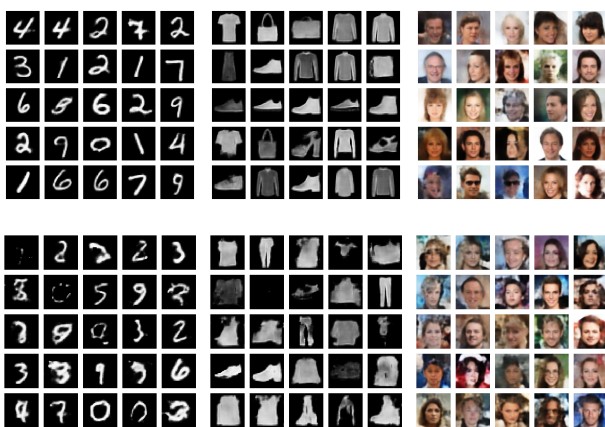

Figure 2: Generated images from DPSWflow-r (upper row) and DPSWgen (lower row) for MNIST, Fashion-MNIST, and Celeba with no DP: $\varepsilon = \infty$.

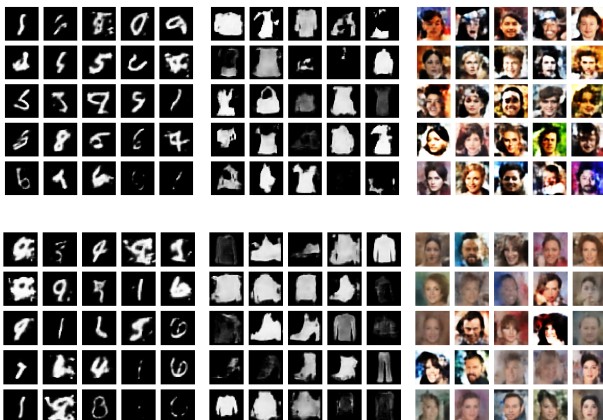

Figure 3: Generated images from DPSWflow-r (upper row) and DPSWgen (lower row) for MNIST, Fashion-MNIST, and Celeba with DP: $\varepsilon = 10$.

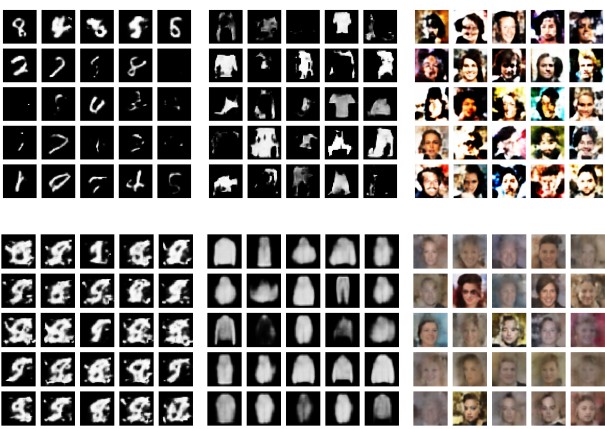

Figure 4: Generated images from DPSWflow-r (upper row) and DPSWgen (lower row) for MNIST, Fashion-MNIST, and Celeba with DP: $\varepsilon = 5$.

resampling (DPSWflow). These results support our consideration of the efficiency of resampling the projections as discussed at the end of Section 3.2. These experiments show that our approach is practically viable and can serve as a promising basis for future work on private generative models.

## 5 Conclusion

In this paper we have introduced a novel theoretically grounded method for differentially private generative modeling. Our approach leverages gradient flows within the Wasserstein space, with a drift term computed using the differentially private sliced Wasserstein distance. To the best of our knowledge, we are the first to propose such a DP gradient flow approach. Our experiments have shown that our approach is practically viable, leading to generated samples of higher quality than those from generator-based models trained with the same DP metric at the same level of privacy. With both a strong theoretical foundation and experimental viability, we believe that our method forms a promising basis for future work in the area of private generative modeling.

### Broader Impact Statement

Some aspects of our contributions in this paper are theoretical in nature, and we do not foresee any adverse societal impacts resulting from them. Our experiments are run on small datasets, and have negligible carbon footprint. Our adherence to differential privacy principles ensures that our generative model aligns with privacy-preserving principles.

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

# A   Proof of Theorem 1

We begin by presenting two propositions that generalize Proposition 5.1.6 and Proposition 5.1.7 of Bonnotte (2013), respectively. These propositions will play a crucial role in the proof of Theorem 1, and constitute a key element of novelty in our proof, compared to the proof of Theorem 2 in Liutkus et al. (2019).

Indeed, the $\mathcal{G}_\sigma \mathcal{SW}^2$ metric is not a simple application of the sliced Wasserstein metric to Gaussian convoluted measures. The convolution with Gaussian measure in the $\mathcal{G}_\sigma \mathcal{SW}_2^2$ metric occurs within the surface integral, separately for each one-dimensional projection of the original measures. To reiterate, $\mathcal{G}_\sigma \mathcal{SW}_2^2(\mu, \nu) \neq \mathcal{SW}_2^2(\mu * \xi_\sigma, \nu * \xi_\sigma)$, where $\xi_\sigma \sim N(0, \sigma)$. This distinction becomes clear when comparing equations 4 with 9. Hence, establishing a DP gradient flow presented a unique challenge which makes a distinct contribution. The distinct nature of $\mathcal{G}_\sigma \mathcal{SW}_2^2$ metric introduces two blockers that need to be circumvented before applying results from Bonnotte (2013) and Liutkus et al. (2019), namely the existence and regularity of minimizers to the functional in Equation 10. Both these steps are non-trivial.

**Proposition 2.** *Let $\mu, \nu \in \mathcal{P}(\Omega)$. For any $\bar{\mu} \in \mathcal{P}(\Omega)$,*

$$\lim_{\varepsilon \to 0^+} \frac{\mathcal{G}_\sigma \mathcal{SW}_2^2((1-\varepsilon)\mu + \varepsilon\bar{\mu}, \nu) - \mathcal{G}_\sigma \mathcal{SW}_2^2(\mu, \nu)}{2\varepsilon} = \fint_{\mathbb{S}^{d-1}} \int_\Omega \psi_{\mu_t,\theta}^{(\sigma)}(\langle\theta, x\rangle) \mathrm{d}(\bar{\mu} - \mu)\mathrm{d}\theta, \qquad (28)$$

*where $\psi_{\mu_t,\theta}^{(\sigma)}$ is a Kantorovich potential between $\theta_\sharp \mu * \mathcal{N}_\sigma$ and $\theta_\sharp \nu * \mathcal{N}_\sigma$.*

*Proof.* Since $\theta_\sharp \nu * \mathcal{N}_\sigma$ is absolutely continuous with respect to the Lebesgue measure for any $\theta \in \mathbb{S}^{d-1}$, there indeed exists a Kantorovich potential $\psi_{\mu_t,\theta}^{(\sigma)}$ between $\theta_\sharp \mu * \mathcal{N}_\sigma$ and $\theta_\sharp \nu * \mathcal{N}_\sigma$. Since $\psi_{\mu_t,\theta}^{(\sigma)}$ may not be optimal between $(1-\varepsilon)\mu + \varepsilon\bar{\mu}$ and $\nu$,

$$\liminf_{\varepsilon \to 0^+} \frac{\mathcal{G}_\sigma \mathcal{SW}_2^2((1-\varepsilon)\mu + \varepsilon\bar{\mu}, \nu) - \mathcal{G}_\sigma \mathcal{SW}_2^2(\mu, \nu)}{2\varepsilon} \geq \fint \int \psi_{\mu_t,\theta}^{(\sigma)}(\langle\theta, x\rangle) \mathrm{d}(\bar{\mu} - \mu)\mathrm{d}\theta. \qquad (29)$$

Conversely, let $\psi_{\mu_t,\theta}^{(\sigma,\varepsilon)}$ be a Kantorovich potential between $\theta_\sharp[(1-\varepsilon)\mu + \varepsilon\bar{\mu}] * \mathcal{N}_\sigma$ and $\theta_\sharp \nu * \mathcal{N}_\sigma$ with $\int \psi_{\mu_t,\theta}^{(\sigma,\varepsilon)} \mathrm{d}\theta_\sharp[(1-\varepsilon)\mu + \varepsilon\bar{\mu}] * \mathcal{N}_\sigma$. Then,

$$\frac{1}{2}\mathcal{G}_\sigma \mathcal{SW}_2^2((1-\varepsilon)\mu + \varepsilon\bar{\mu}, \nu) - \frac{1}{2}\mathcal{G}_\sigma \mathcal{SW}_2^2(\mu, \nu) \leq \varepsilon \fint \int \psi_{\mu_t,\theta}^{(\sigma,\varepsilon)}(\langle\theta, x\rangle) \mathrm{d}(\bar{\mu} - \mu)\mathrm{d}\theta. \qquad (30)$$

As in the proof of Proposition 5.1.6 in Bonnotte (2013), $\psi_{\mu_t,\theta}^{(\sigma,\varepsilon)}$ uniformly converges to a Kantorovich potential for $(\theta_\sharp \mu * \mathcal{N}_\sigma, \theta_\sharp \nu * \mathcal{N}_\sigma)$ as $\varepsilon \to 0^+$. Hence,

$$\limsup_{\varepsilon \to 0^+} \frac{\mathcal{G}_\sigma \mathcal{SW}_2^2((1-\varepsilon)\mu + \varepsilon\bar{\mu}, \nu) - \mathcal{G}_\sigma \mathcal{SW}_2^2(\mu, \nu)}{2\varepsilon} \leq \fint \int \psi_{\mu_t,\theta}^{(\sigma,\varepsilon)}(\langle\theta, x\rangle) \mathrm{d}(\bar{\mu} - \mu)\mathrm{d}\theta. \qquad (31)$$

Combining Eq. (30) and Eq. (31), we get the desired result. $\qquad \square$

**Proposition 3.** *Let $\mu, \nu \in \mathcal{P}(\Omega)$. For any diffeomorphism $\zeta$ of $\Omega$,*

$$\lim_{\varepsilon \to 0^+} \frac{\mathcal{G}_\sigma \mathcal{SW}_2^2([\mathrm{Id} + \varepsilon\zeta]_\sharp \mu, \nu) - \mathcal{G}_\sigma \mathcal{SW}_2^2(\mu, \nu)}{2\varepsilon} = \fint_{\mathbb{S}^{d-1}} \int_\Omega (\psi_{\mu_t,\theta}^{(\sigma)})'(\langle\theta, x\rangle)\langle\theta, \zeta(x)\rangle \mathrm{d}\mu\mathrm{d}\theta, \qquad (32)$$

*where $\psi_{\mu_t,\theta}^{(\sigma)}$ is a Kantorovich potential between $\theta_\sharp \mu * \mathcal{N}_\sigma$ and $\theta_\sharp \nu * \mathcal{N}_\sigma$.*

*Proof.* Using the fact that $\psi_{\mu_t,\theta}^{(\sigma)}$ is a Kantorovich potential between $\theta_\sharp \mu * \mathcal{N}_\sigma$ and $\theta_\sharp \nu * \mathcal{N}_\sigma$, we get the following:

$$\frac{\mathcal{G}_\sigma \mathcal{SW}_2^2([\mathrm{Id} + \varepsilon\zeta]_\sharp \mu, \nu) - \mathcal{G}_\sigma \mathcal{SW}_2^2(\mu, \nu)}{2\varepsilon} \geq \fint_{\mathbb{S}^{d-1}} \int_\Omega \frac{\psi_{\mu_t,\theta}^{(\sigma)}(\langle\theta, x + \varepsilon\zeta(x)\rangle) - \psi_{\mu_t,\theta}^{(\sigma)}(\langle\theta, x\rangle)}{2\varepsilon} \mathrm{d}\mu\mathrm{d}\theta. \qquad (33)$$

Since the Kantorovich potential is Lipschitz, it is differentiable almost everywhere, and so, we get the following using Lebesgue's differentiation theorem:

$$\liminf_{\varepsilon \to 0^+} \frac{\mathcal{G}_\sigma \mathcal{SW}_2^2([\mathrm{Id} + \varepsilon\zeta]_\sharp\mu, \nu) - \mathcal{G}_\sigma \mathcal{SW}_2^2(\mu, \nu)}{2\varepsilon} \geq \fint_{\mathbb{S}^{d-1}} \int_\Omega (\psi_{\mu_t,\theta}^{(\sigma)})'(\langle \theta, x\rangle)\langle \theta, \zeta(x)\rangle \mathrm{d}\mu \mathrm{d}\theta. \tag{34}$$

Conversely, we will now show the same upper bound on the lim sup. Let $\gamma_{\theta,\sigma} \in \Pi(\theta_\sharp\mu * \mathcal{N}_\sigma, \theta_\sharp\nu * \mathcal{N}_\sigma)$ be the optimal transport plan corresponding to $\psi_{\mu_t,\theta}^{(\sigma)}$. As is done in Proposition 5.1.7 in Bonnotte (2013), we extend $\gamma_{\theta,\sigma}$ to $\pi_{\theta,\sigma} \in \Pi(\mu, \nu)$ such that $(\theta \otimes \theta)_\sharp\pi_{\theta,\sigma} = \gamma_{\theta,\sigma}$. In other words, if random variables $(X, Y)$ are sampled from $\pi_{\theta,\sigma}$, then $(\langle X, \theta\rangle, \langle Y, \theta\rangle)$ will follow the law of $\gamma_{\theta,\sigma}$ for every $\theta \in \mathbb{S}^{d-1}$. Then, it follows that $[\theta + \varepsilon\theta(\zeta) \otimes \theta]_\sharp\pi_\theta \in \Pi(\theta_\sharp[\mathrm{Id} + \varepsilon\zeta]_\sharp\mu, \theta_\sharp\nu)$. Hence,

$$\mathcal{G}_\sigma \mathcal{SW}_2^2([\mathrm{Id} + \varepsilon\zeta]_\sharp\mu, \nu) - \mathcal{G}_\sigma \mathcal{SW}_2^2(\mu, \nu) \leq \fint \int |\langle \theta, x + \varepsilon\zeta(x) - y\rangle|^2 - |\langle \theta, x - y\rangle|^2 \mathrm{d}\pi_{\theta,\sigma}(x, y)\mathrm{d}\theta.$$

Since $\pi_{\theta,\sigma}$ is constructed from $\gamma_{\theta,\sigma}$, which in turn is based on the Kantorovich potential $\psi_{\mu_t,\theta}^{(\sigma)}$, we have $\langle \theta, y\rangle = \langle \theta, x\rangle - (\psi_{\mu_t,\theta}^{(\sigma)})'(\langle \theta, x\rangle)$ for $\pi_{\theta,\sigma}$-a.e. $(x, y)$, because of the optimality of $\gamma_{\theta,\sigma}$ for the one-dimensional optimal transport. Therefore,

$$\mathcal{G}_\sigma \mathcal{SW}_2^2([\mathrm{Id} + \varepsilon\zeta]_\sharp\mu, \nu) - \mathcal{G}_\sigma \mathcal{SW}_2^2(\mu, \nu) \leq \fint \int |(\psi_{\mu_t,\theta}^{(\sigma)})'(\langle \theta, x\rangle) - \varepsilon\langle \theta, \zeta(x)\rangle|^2 - |(\psi_{\mu_t,\theta}^{(\sigma)})'(\langle \theta, x\rangle)|^2 \mathrm{d}\pi_{\theta,\sigma}(x, y)\mathrm{d}\theta.$$

Simplifying the right hand side of the above equation and taking the limit of $\varepsilon \to 0^+$, we get the following:

$$\limsup_{\varepsilon \to 0^+} \frac{\mathcal{G}_\sigma \mathcal{SW}_2^2([\mathrm{Id} + \varepsilon\zeta]_\sharp\mu, \nu) - \mathcal{G}_\sigma \mathcal{SW}_2^2(\mu, \nu)}{2\varepsilon} \leq \fint_{\mathbb{S}^{d-1}} \int_\Omega (\psi_{\mu_t,\theta}^{(\sigma)})'(\langle \theta, x\rangle)\langle \theta, \zeta(x)\rangle \mathrm{d}\mu \mathrm{d}\theta. \tag{35}$$

Combining Eq. (35) and Eq. (34), we get the desired result. $\qquad\square$

We reproduce the following definition from (Liutkus et al., 2019).

**Definition 2.** *(Generalized Minimizing Movement Scheme (GMMS)) Let $r > 0$ and $\mathcal{F}: \mathbb{R}_+ \times \mathcal{P}(\overline{\mathrm{B}}(0, r)) \times \mathcal{P}(\overline{\mathrm{B}}(0, r)) \to \mathbb{R}$ be a functional. For $h > 0$, let $\mu^h: [0, \infty) \to \mathcal{P}(\overline{\mathrm{B}}(0, r))$ be a piecewise constant trajectory for $\mathcal{F}$ starting at $\mu_0 \in \mathcal{P}(\overline{\mathrm{B}}(0, r))$, such that: (i) $\mu^h(0) = \mu_0$, (ii) $\mu^h(t) = \mu^h(nh)$ for $n = \lfloor t/h \rfloor$, and (iii) $\mu^h((n+1)h)$ minimizes the functional $\zeta \mapsto \mathcal{F}(h, \zeta, \mu^h(nh))$, for all $n \in \mathbb{N}$.*

*We say that $\hat{\mu}$ is a Minimizing Movement Scheme (MMS) for $\mathcal{F}$ starting at $\mu_0$ if there exists a family of piecewise constant trajectories $(\mu^h)_{h>0}$ for $\mathcal{F}$ such that $\lim_{h \to 0} \mu^h(t) = \hat{\mu}(t)$ for all $t \in \mathbb{R}_+$.*

*We say that $\tilde{\mu}$ is a Generalized Minimizing Movement Scheme (GMMS) for $\mathcal{F}$ starting at $\mu_0$ if there exists a family of piecewise constant trajectories $(\mu^{h_n})_{n \in \mathbb{N}}$ for $\mathcal{F}$ such that $\lim_{n \to \infty} \mu^{h_n}(t) = \tilde{\mu}(t)$ for all $t \in \mathbb{R}_+$.*

**Theorem 4** (Existence of solution to the minimization functional). *Let $\nu \in \mathcal{P}(\overline{\mathrm{B}}(0, 1))$ and $r > \sqrt{d}$. For any $\mu_0 \in \mathcal{P}(\overline{\mathrm{B}}(0, r))$ with a density $\rho_0 \in \mathrm{L}^\infty(\overline{\mathrm{B}}(0, r))$ and $h > 0$, there exists a $\hat{\mu} \in \mathcal{P}(\overline{\mathrm{B}}(0, r))$ that minimizes the following functional:*

$$\mathcal{G}(\mu) = \mathcal{F}_{\lambda,\sigma}^\nu(\mu) + \frac{1}{2h}\mathcal{W}_2^2(\mu, \mu_0), \tag{36}$$

*where $\mathcal{F}_{\lambda,\sigma}^\nu(\mu)$ is given by Eq. (10). Moreover $\hat{\mu}$ admits a density $\hat{\rho}$ on $\overline{\mathrm{B}}(0, r)$.*

*Proof.* We note that $\mathcal{P}(\overline{\mathrm{B}}(0, 1))$ is compact for weak convergence (and equivalently for convergence in $W_2$). Hence, showing that $\mathcal{G}(\mu)$ is lower semi-continuous on $\mathcal{P}(\overline{\mathrm{B}}(0, 1))$ would suffice to show the existence of a solution $\hat{\mu}$. By Lemma 9.4.3 of Ambrosio (2008), $\mathcal{H}$ is lower semi-continuous. By Rakotomamonjy & Ralaivola (2021), $\mathcal{G}_\sigma \mathcal{SW}_2(\mu, \nu)$ is symmetric and satisfies the triangle inequality. Moreover, $\mathcal{G}_\sigma \mathcal{SW}_2(\mu, \nu) \leq \mathcal{SW}_2(\mu, \nu)$ for any $\sigma \geq 0$. Hence for any $\xi, \xi' \in \mathcal{P}(\overline{\mathrm{B}}(0, 1))$,

$$|\mathcal{G}_\sigma \mathcal{SW}_2(\xi, \nu) - \mathcal{G}_\sigma \mathcal{SW}_2(\xi', \nu)| \leq \mathcal{G}_\sigma \mathcal{SW}_2(\xi, \xi') \leq \mathcal{SW}_2(\xi, \xi') \leq c_d \mathcal{W}(\xi, \xi'),$$

where $c_d > 0$ is a constant only dependent on the dimension $d$, and the last inequality follows from Proposition 5.1.3 in Bonnotte (2013). Hence, there exists a minimum $\hat{\mu} \in \mathcal{P}(\overline{\mathrm{B}}(0, r))$ of $\mathcal{G}(\mu)$. Moreover, $\hat{\mu}$ must admit a density $\hat{\rho}$ because otherwise $\mathcal{H}(\hat{\mu}) = \infty$. $\qquad\square$

**Lemma 3** (Regularity of the solution to the minimizing functional). *Under the assumptions of Theorem 4, any minimizer $\hat{\mu}$ of $\mathcal{G}(\mu)$ in Eq. (36) must admit a strictly positive density $\hat{\rho} > 0$ a.e., and $\|\hat{\rho}\|_{L^\infty} \leq (1 + h/\sqrt{d})^{\sqrt{d}}\|\rho_0\|_{L^\infty}$.*

*Proof.* By Theorem 4, a minimizer $\hat{\mu}$ of $\mathcal{G}(\mu)$ exists and admits a density $\hat{\rho}$. Let $\bar{\mu} \in \mathcal{P}(\overline{B}(0,1))$ be an arbitrary probability measure with density $\bar{\rho}$. For $\varepsilon \in (0,1)$ let $\rho_\varepsilon = (1-\varepsilon)\hat{\rho} + \varepsilon\bar{\rho}$ and let $\mu_\varepsilon \in \mathcal{P}(\overline{B}(0,1))$ be the probability measure corresponding to the density $\rho_\varepsilon$. By the optimality of $\hat{\rho}$, we have that $\mathcal{G}(\hat{\mu}) \leq \mathcal{G}(\mu_\varepsilon)$. Hence,

$$0 \geq \lim_{\varepsilon \to 0^+} \frac{\mathcal{G}(\hat{\mu}) - \mathcal{G}(\mu_\varepsilon)}{\varepsilon} = \lim_{\varepsilon \to 0^+} \frac{\mathcal{G}_\sigma \mathcal{SW}_2^2(\hat{\mu}) - \mathcal{G}_\sigma \mathcal{SW}_2^2(\mu_\varepsilon)}{2\varepsilon} + \lambda \limsup_{\varepsilon \to 0^+} \frac{\mathcal{H}(\hat{\mu}) - \mathcal{H}(\mu_\varepsilon)}{\varepsilon} + \lim_{\varepsilon \to 0^+} \frac{\mathcal{W}_2^2(\hat{\mu}) - \mathcal{W}_2^2(\mu_\varepsilon)}{2h\varepsilon}$$

$$= \fint_{\mathbb{S}^{d-1}} \int_\Omega \psi_{\mu_t,\theta}^{(\sigma)}(\langle \theta, x \rangle)\mathrm{d}(\bar{\mu} - \hat{\mu})\mathrm{d}\theta + \lambda \limsup_{\varepsilon \to 0^+} \frac{\mathcal{H}(\hat{\mu}) - \mathcal{H}(\mu_\varepsilon)}{\varepsilon} + \int_\Omega \phi \mathrm{d}(\bar{\mu} - \hat{\mu}),$$

where the last equality follows by combining Proposition 2 with Proposition 1.5.6 in (Bonnotte, 2013). Here, $\psi_{\mu_t,\theta}^{(\sigma)}$ is a Kantorovich potential between $\theta_\sharp \hat{\mu} * \mathcal{N}_\sigma$, and $\theta_\sharp \nu * \mathcal{N}_\sigma$ as in Proposition 2 and $\phi$ is a Kantorovich potential between $\hat{\mu}$ and $\nu$ for $\mathcal{W}_2$. Rearranging, we get the following:

$$\limsup_{\varepsilon \to 0^+} \frac{\mathcal{H}(\hat{\mu}) - \mathcal{H}(\mu_\varepsilon)}{\varepsilon} \leq \frac{1}{\lambda} \int_{\overline{B}(0,r)} \Psi \mathrm{d}(\bar{\mu} - \hat{\mu}), \tag{37}$$

where $\Psi(x) := \fint_{\mathbb{S}^{d-1}} \psi_{\mu_t,\theta}^{(\sigma)}(\langle \theta, x \rangle) + \frac{1}{h}\phi(x)$. From this point, for any $\mu_0 \in \mathcal{P}(\overline{B}(0,r))$ with a density $\rho_0$ that is smooth and strictly positive, we get the desired result by following the proof strategy of Lemma 5.4.3 in (Bonnotte, 2013). For a more general $\mu_0$ with a density $\rho_0 \in L^\infty(\overline{B}(0,r))$, we again arrive at the desired result by following the proof strategy of Theorem S4 in (Liutkus et al., 2019), which proceeds by smoothing $\rho_0$ by convolution with a Gaussian. $\square$

**Theorem 5** (Existence of GMMS). *Under the assumptions of Theorem 4, there exists a GMMS $(\mu_t)_{t \geq 0}$ in $\mathcal{P}(\overline{B}(0,r))$, starting from $\mu_0$ for the following functional:*

$$\mathcal{F}_{\lambda,\sigma}^\nu(h, \mu_{nxt}, \mu_{prv}) = \mathcal{F}_{\lambda,\sigma}^\nu(\mu_{nxt}) + \frac{1}{2h}\mathcal{W}_2^2(\mu_{nxt}, \mu_{prv}). \tag{38}$$

*Moreover, for any $t > 0$, $\mu_t$ has a density $\rho_t$ such that $\|\rho_t\|_{L^\infty} \leq e^{td\sqrt{d}}\|\rho_0\|_{L^\infty}$.*

*Proof.* The desired result follows straightforwardly by following the proof of Theorem S5 in Liutkus et al. (2019) or Theorem 5.5.3 in Bonnotte (2013), with the support of Lemma 3 and Theorem 4. $\square$

**Theorem 6** (Continuity equation for GMMS). *Under the assumptions of Theorem 4, let $(\mu_t)_{t \geq 0}$ be the GMMS given by Theorem 5. For $\theta \in \mathbb{S}^{d-1}$, let $\psi_{\mu_t,\theta}^{(\sigma)}$ be the Kantorovich potential between $P_\#^\theta \mu_t * \xi_\sigma$ and $P_\#^\theta \nu * \xi_\sigma$, with $\xi_\sigma \sim \mathcal{N}(0, \sigma^2)$. For $t \geq 0$, the density $\rho_t$ of $\mu_t$ satisfies the following continuity equation in a weak sense:*

$$\frac{\partial \rho_t}{\partial t} = -\mathrm{div}(v_t^{(\sigma)}\rho_t) + \lambda\Delta\rho_t,$$

*with:*

$$v_t^{(\sigma)}(x) = v^{(\sigma)}(x, \mu_t) = \int_{\mathbb{S}^{d-1}} (\psi_{\mu_t,\theta}^{(\sigma)})'(\langle x, \theta \rangle)\theta \mathrm{d}\theta.$$

*That is, for all $\xi \in C_c^\infty([0,\infty) \times \overline{B}(0,r))$,*

$$\int_0^\infty \int_{\overline{B}(0,r))} \left[\frac{\partial \xi}{\partial t}(t,x) - v_t^{(\sigma)}\nabla\xi(t,x) - \lambda\Delta\xi(t,x)\right]\rho_t(x)\mathrm{d}x\mathrm{d}t = -\int_{\overline{B}(0,r))} \xi(0,x)\rho_0(x)\mathrm{d}x. \tag{39}$$

*Proof.* We will closely follow the proof of Theorem S6 in Liutkus et al. (2019) and Theorem 5.6.1 in Bonnotte (2013). Just as in the proof of Theorem S6 in Liutkus et al. (2019), we will proceed in five steps.

*Step (1):* By the definition of GMMS, there exists a family of piecewise constant trajectories $(\mu^{h_n})_{n \in \mathbb{N}}$ for $\mathcal{F}^\nu_{\lambda,\sigma}$ such that $\lim_{n \to \infty} \mu^{h_n}_t = \mu_t$ for all $t \in \mathbb{R}_+$. Let $\xi \in C^\infty_c([0,\infty) \times \overline{B}(0,r))$ and let $\xi^n_k(x)$ denote $\xi(kh_n, x)$. Using step 1 of the proof of Theorem S6 in Liutkus et al. (2019), we get:

$$\fint_{\overline{B}(0,r)} \xi(0,x)\rho_0(x)\mathrm{d}x + \int_0^\infty \fint_{\overline{B}(0,r)} \frac{\partial \xi}{\partial t}(t,x)\rho_t(x)\mathrm{d}x\mathrm{d}t = \lim_{n \to \infty} -h_n \sum_{k=1}^\infty \fint_{\overline{B}(0,r)} \xi^n_k(x) \frac{\rho^{h_n}_{kh_n}(x) - \rho^{h_n}_{(k-1)h_n}(x)}{h_n}\mathrm{d}x. \tag{40}$$

*Step (2):* For any $\theta \in \mathbb{S}^{d-1}$, let $\psi^{(\sigma),h_n}_{\mu_t,\theta}$ be the Kantorovich potential between $P^\theta_\# \mu^{h_n}_t * \xi_\sigma$ and $P^\theta_\# \nu * \xi_\sigma$, with $\xi_\sigma \sim \mathcal{N}(0,\sigma^2)$. Using step 2 of the proof of Theorem S6 in Liutkus et al. (2019), we get:

$$\int_0^\infty \fint_{\overline{B}(0,r)} \fint (\psi^{(\sigma)}_{\mu_t,\theta})'(\langle\theta,x\rangle)\langle\theta,\nabla\xi(x,t)\rangle\mathrm{d}\theta\mathrm{d}\mu_t(x)\mathrm{d}t = \lim_{n \to \infty} h_n \sum_{k=1}^\infty \fint_{\overline{B}(0,r)} \fint \psi^{(\sigma),h_n}_{\mu^{h_n}_k,\theta}(\theta^*)\langle\theta,\nabla\xi^n_k\rangle\mathrm{d}\theta\mathrm{d}\mu^{h_n}_{kh_n}. \tag{41}$$

*Step (3):* From step 3 of the proof of Theorem S6 in Liutkus et al. (2019), we get:

$$\lim_{n \to \infty} h_n \sum_{k=1}^\infty \fint_{\overline{B}(0,r)} \Delta\xi^n_k(x)\rho^{h_n}_{kh_n}(x)\mathrm{d}x = \int_0^\infty \fint_{\overline{B}(0,r)} \Delta\xi(t,x)\rho_t(x)\mathrm{d}x\mathrm{d}t. \tag{42}$$

*Step (4):* Let $\phi^{h_n}_k$ be the Kantorovich potential from $\mu^{h_n}_{kh_n}$ to $\mu^{h_n}_{(k-1)h_n}$. From the optimality of $\mu^{h_n}_{kh_n}$, the first variation of the functional $\zeta \mapsto \mathcal{F}^\nu_{\lambda,\sigma}(\zeta) + \frac{1}{2h}\mathcal{W}^2_2(\zeta, \mu^{h_n}_{kh_n})$ with respect to $\zeta$ at the point $\mu^{h_n}_{kh_n}$ in the direction of the vector field $\nabla\xi^n_k$ is zero. Using Proposition 3 for the first variation of the $\mathcal{G}_\sigma \mathcal{SW}^2_2$ term, Proposition 5.1.7 for the first variation of the $\mathcal{W}^2_2$ term, and Jordan et al. (1998) for the first variation of the $\mathcal{H}$ term, we get the following:

$$0 = \frac{1}{h_n} \int_{\overline{B}(0,r)} \langle\nabla\phi^{h_n}_k(x), \nabla\xi^n_k(x)\rangle\mathrm{d}\mu^{h_n}_{kh_n}(x) - \int_{\overline{B}(0,r)} \fint (\psi^{(\sigma),h_n}_{\mu^{h_n}_k,\theta})'(\theta^*)\langle\theta,\nabla\xi^n_k(x)\rangle\mathrm{d}\theta\mathrm{d}\mu^{h_n}_{kh_n}(x)$$
$$- \lambda \int_{\overline{B}(0,r)} \Delta\xi^n_k(x)\mathrm{d}\mu^{h_n}_{kh_n}(x). \tag{43}$$

Proceeding as in step 4 of Liutkus et al. (2019) and using Eq. (43), we get the following:

$$\lim_{n \to \infty} -h_n \sum_{k=1}^\infty \xi^n_k(x) \frac{\rho^{h_n}_{kh_n} - \rho^{h_n}_{(k-1)h_n}}{h_n}\mathrm{d}x$$
$$= \lim_{n \to \infty} \left( h_n \sum_{k=1}^\infty \int_{\overline{B}(0,r)} \fint (\psi^{(\sigma),h_n}_{\mu^{h_n}_k,\theta})'(\theta^*)\langle\theta,\nabla\xi^n_k\rangle\mathrm{d}\theta\mathrm{d}\mu^{h_n}_{kh_n} + h_n \sum_{k=1}^\infty \int_{\overline{B}(0,r)} \Delta\xi^n_k(x)\rho^{h_n}_{kh_n}(x)\mathrm{d}x \right). \tag{44}$$

*Step (4):* Combining Eq. (40), Eq. (41), Eq. (42) and Eq. (44), we get the desired result in Eq. (39).

$\square$

# B    Proof of Theorem 2

We simply follow the proof strategy of Theorem 3 in Liutkus et al. (2019). We begin by restating the following two discrete-time SDEs:

$$\widehat{X}_{k+1} = hv^{(\sigma)}(\widehat{X}_k, \widehat{\mu}_{kh}) + \sqrt{2\lambda h}Z_{k+1}, \tag{45}$$
$$\bar{X}_{k+1} = h\hat{v}^{(\sigma)}(\bar{X}_k, \bar{\mu}_{kh}) + \sqrt{2\lambda h}Z_{k+1}, \tag{46}$$

where Eq. (45) is equivalent to Eq. (17), which in turn is the Euler-Maruyama discretization of the continuous-time SDE in Eq. (6). Equation 46 is equivalent to the particle update equation in Eq. (18).

Similar to the proof of Theorem 3 in Liutkus et al. (2019), we define two continuous-time processes $(Y_t)_{t \geq 0}$ and $(U_t)_{t \geq 0}$, defined by the following continuous-time SDEs:

$$dY_t = \tilde{v}_t^{(\sigma)}(Y)dt + \sqrt{2\lambda}dW_t, \tag{47}$$

$$dU_t = \bar{v}_t^{(\sigma)}(U)dt + \sqrt{2\lambda}dW_t, \tag{48}$$

where

$$\tilde{v}_t(Y) := -\sum_{k=0}^{\infty} \hat{v}_{kh}^{(\sigma)}(Y_{kh}, \hat{\mu}_{kh})\mathbb{1}_{[kh,(k+1)h)}(t), \tag{49}$$

$$\tilde{v}_t(U) := -\sum_{k=0}^{\infty} \hat{v}_{kh}^{(\sigma)}(U_{kh}, \bar{\mu}_{kh})\mathbb{1}_{[kh,(k+1)h)}(t). \tag{50}$$

In Eq. (49), $\hat{\mu}_{kh}$ follows the distribution of $\widehat{X}_k$ in the discrete-time process defined by the update equation in Eq. (45). Therefore $(Y_t)_{t \geq 0}$ is a continuous linear interpolation of the discrete-time process $(\widehat{X}_k)_{k \in \mathbb{N}_+}$.

In Eq. (50), $\bar{\mu}_{kh}$ follows the distribution of $\bar{X}_k$ in the discrete-time process defined by the update equation in Eq. (46). Therefore $(U_t)_{t \geq 0}$ is a continuous linear interpolation of the discrete-time process $(\bar{X}_k)_{k \in \mathbb{N}_+}$.

Let $\pi_X^T$, $\pi_Y^T$, and $\pi_U^T$ denote the distributions of $(X_t)_{t \in [0,T]}$, $(Y_t)_{t \in [0,T]}$, and $(U_t)_{t \in [0,T]}$, respectively, with $T = Kh$. We have the following lemma bounding the total variation distance between the pairs $(\pi_X^T, \pi_Y^T)$ and $(\pi_Y^T, \pi_U^T)$ from Liutkus et al. (2019):

**Lemma 4** (Lemmas S1 and S2 in Liutkus et al. (2019)). *For all $\lambda > 0$, assume that the continuous-time SDE in Eq. (6) has a unique strong solution $(X_t)_{t \geq 0}$ for any starting point $x \in \mathbb{R}^d$. For $t \geq 0$, define $\Psi_{\mu_t,\theta}^{(\sigma)}(x) := \fint_{\mathbb{S}^{d-1}} \psi_{\mu_t,\theta}^{(\sigma)}(\langle \theta, x \rangle)d\theta$. Suppose there exists constants $A, B, L, m, b > 0$, and $\kappa \in (0,1)$, such that the following are true for any $x, x' \in \mathbb{R}^d$, $\mu, \mu' \in \mathcal{P}(\Omega)$, and all $t \geq 0$:*

$$\|v_t^{(\sigma)}(x) - v_{t'}^{(\sigma)}(x')\| \leq L(\|x - x'\| + |t - t'|),$$

$$\|\hat{v}^{(\sigma)}(x, \mu) - \hat{v}^{(\sigma)}(x', \mu')\| \leq L(\|x - x'\| + \|\mu - \mu'\|_{\text{TV}}),$$

$$\langle x, v_t^{(\sigma)}(x) \rangle \geq m\|x\|^2 - b,$$

$$\mathbb{E}[\hat{v}_t^{(\sigma)}] = v_t^{(\sigma)},$$

$$\mathbb{E}[\|\hat{v}^{(\sigma)}(x, \mu_t) - v^{(\sigma)}(x, \mu_t)\|^2] \leq 2\kappa(L^2\|x\|^2 + B^2).$$

*Define:*

$$C_e := \mathcal{H}(\mu_0),$$

$$C_0 := C_e + 2(1 \vee \frac{1}{m})(b + 2B^2 + d\lambda),$$

$$C_1 := 12(L^2 C_0 + B^2) + 1,$$

$$C_2 := 2(L^2 C_0 + B^2).$$

*Then, we have:*

$$\|\pi_X^T - \pi_Y^T\|_{\text{TV}}^2 \leq \frac{L^2 K}{4\lambda}\left(\frac{C_1 h^3}{3} + 3\lambda d h^2\right) + \frac{C_2 \kappa K h}{8\lambda},$$

$$\|\pi_Y^T - \pi_U^T\|_{\text{TV}}^2 \leq \frac{L^2 K h}{16\lambda}\|\pi_X^T - \pi_U^T\|_{\text{TV}}^2.$$

**Theorem 7.** *Under the assumptions of Lemma 4 and for $\lambda > TL^2/8$:*

$$\|\mu_T - \widehat{\mu}_{Kh}\|_{TV}^2 \leq \frac{T}{\lambda - TL^2/8}\left[L^2 h(c_1 h + d\lambda) + c_2\kappa\right], \tag{51}$$

*where $c_1, c_2, L, \kappa > 0$ are constants independent of time.*

*Proof.* We will emulate the proof of Theorem 3 in Liutkus et al. (2019).

$$\begin{aligned}
\|\pi_X^T - \pi_U^T\|_{\mathrm{TV}}^2 &\leq 2\|\pi_X^T - \pi_Y^T\|_{\mathrm{TV}}^2 + 2\|\pi_Y^T - \pi_U^T\|_{\mathrm{TV}}^2 \\
&\leq \frac{L^2 K}{2\lambda}\left(\frac{C_1 h^3}{3} + 3\lambda dh^2\right) + \frac{C_2\kappa Kh}{4\lambda} + \frac{L^2 Kh}{8\lambda}\|\pi_X^T - \pi_U^T\|_{\mathrm{TV}}^2 \\
&\leq \left(1 - \frac{KL^2 h}{8\lambda}\right)^{-1}\left\{\frac{L^2 K}{2\lambda}\left(\frac{C_1 h^3}{3} + 3\lambda dh^2\right) + \frac{C_2\kappa Kh}{4\lambda}\right\},
\end{aligned}$$

where the second inequality follows from Lemma 4 and the last inequality holds for $\lambda > TL^2/8$. The desired result then follows by plugging in $T = Kh$ and rearranging. $\square$

## C  Experimental Details

In this section, we explain all experimental details required running the experiments (along with the code which is provided in the supplementary material). For this project we use 1 NVIDIA GPU Tesla V100 which was necessary for the pretraining of the auto-encoder only. All neural networks are coded using PyTorch (Paszke et al., 2019).

### C.1  Datasets and Evaluation Metric

**MNIST** is a standard dataset introduced in LeCun et al. (1998), with no clear license to the best of our knowledge, composed of monochrome images of hand-written digits. Each MNIST image is single-channel, of size $28 \times 28$. We preprocess MNIST images by extending them to $32 \times 32$ frames (padding each image with black pixels), in order to better fit as inputs and outputs of standard convolutional networks. MNIST is comprised of a training and testing dataset, but no validation set. We split the training set into two equally-sized public and private sets.

**FashionMNIST** is a similar dataset introduced by Xiao et al. (2017) under the MIT license, composed of fashion-related images. Each FashionMNIST image is single-channel, of size 28 x 28, and we also preprocess them by extending them to $32 \times 32$ frames. Like MNIST, FashionMNIST is comprised of a training and testing dataset, but no validation set. We split the training set into two equally-sized public and private sets.

Notice that for MNIST and FashionMNIST, we scaled the pixel values to [0,1].

**CelebA** is a dataset composed of celebrity pictures introduced by (Liu et al., 2015). Its license permits use for non-commercial research purposes. Each CelebA image has three color channels, and is of size $178 \times 218$. We preprocess these images by center-cropping each to a square image and resizing to $64 \times 64$. CelebA is comprised of a training, testing, and validation dataset. After conducting initial experiments and analysis with the validation set, we removed it. We then split the training set into two equally-sized public and private datasets.

**Fréchet Inception Distance (FID)** was introduced by Heusel et al. (2018). It measures the generative performance of the models we consider. In our code we use the PyTorch implementation of TorchMetrics (Skafte Detlefsen et al., 2022).

### C.2  Structure of the Autoencoders for Data Dimension Reduction

The experiments performed on MNIST and FashionMNIST datasets utilize an autoencoder architecture as per the framework proposed by Liutkus et al. (2019). Furthermore, we normalized the latent space before

adding Gaussian noise, ensuring that the encoded representations lie on a hyper-sphere. We use the following autoencoder structure, which is the same as that used in Liutkus et al. (2019):

- **Encoder.** Four 2D convolution layers with (num chan out, kernel size, stride, padding) set to (3, 3, 1, 1), (32, 4, 2, 0), (32, 3, 1, 1), (32, 3, 1, 1), each one followed by a ReLU activation. At the output, a linear layer is set to the desired bottleneck size, and then the outputs are normalized.

- **Decoder.** A linear layer receives from the bottleneck features a vector of dimension 8192, which is reshaped as (32, 16, 16). Then, three convolution layers are applied, all with 32 output channels and (kernel size, stride, panning) set to, respectively, (3, 1, 1), (3, 1, 1), (2, 2, 0). A 2D convolution layer is then applied, with the specified output number of channels set to that of the data (1 for black and white, 3 for color), and a (kernel size, stride, panning) set to (3, 1, 1). All layers are followed by a ReLU activation, and a sigmoid activation is applied to the final output.

Conversely, experiments conducted on the CelebA dataset employ an autoencoder/generator architecture based on the DCGAN framework proposed by Radford et al. (2016a). The structure is the following:

- **Encoder.** Four 2D convolution layers are employed with the following specifications: $(64,3,1,1)$, $(64\times2,4,2,1)$, $(64\times4,4,2,1)$ and $(64\times8,4,2,1)$. Each convolutional layer is followed by a leaky Rectified Linear Unit (LeakyReLU) activation function. Subsequently, a linear layer is applied to obtain the desired bottleneck size. The outputs are then normalized.

- **Decoder.** Four 2D convolution layers are employed with the following specifications: $(64\times8,4,1,0)$, $(64\times4,4,2,1)$, $(64\times2,4,2,1)$ and $(64,4,2,1)$. Each convolutional layer is followed by a leaky Rectified Linear Unit (LeakyReLU) activation function and batch normalization is added after the activation.

## C.3   Additional Algorithm

Algorithm 2 describes the DP Sliced Wasserstein Flow without resampling of the $\theta$s.

---

**Algorithm 2:** DP Sliced Wasserstein Flow without resampling of the $\theta$s: DPSWflow.

---

1: **Input:** $Y = [y_1^T, \ldots, y_n^T]^T \in \mathbb{R}^{n \times d}$ i.e. $N$ i.i.d. samples from target distribution $\nu$, number of projections $N_\theta$, regularization parameter $\lambda$, variance $\sigma$, step size $h$.
2: **Output:** $X = [x_1^T, \ldots, x_n^T]^T \in \mathbb{R}^{n \times d}$
3: *// Initialize the particles*
4: $\{x_i\}_{i=1}^N \sim \mu_0$, $\widehat{X} = [x_1^T, \ldots, x_n^T]^T \in \mathbb{R}^{n \times d}$
5: $\{\theta_j\}_{j=1}^{N_\theta} \sim \mathrm{Unif}(\mathbb{S}^{d-1})$, $\Theta = [\theta_1^T, \ldots, \theta_{N_\theta}^T]$
6: Sample $Z_Y \in \mathbb{R}^{n \times N_\theta}$ from i.i.d. $\mathcal{N}(0, \sigma^2)$
7: Compute the inverse CDF of $Y\Theta + Z_Y$.
8: *// Iterations*
9: **for** $k = 0, \ldots, K-1$ **do**
10:    Sample $M_\theta$ projections among $\Theta$ to obtain $\Upsilon$
11:    Sample $Z_X \in \mathbb{R}^{n \times M_\theta}$ from i.i.d. $\mathcal{N}(0, \sigma^2)$
12:    Compute the CDF of $X\Upsilon + Z_X$
13:    Compute $\widehat{v}^{(\sigma)}(x_i)$ using Eq. (20)
14:    $x_i \leftarrow h\widehat{v}^{(\sigma)}(x_i) + \sqrt{2\lambda h}z$, $z \sim \mathcal{N}(0, \mathcal{I}_d)$
15: **end for**

---

## C.4   Understanding Performance Differences Between DPSWflow-r and DPSWflow.

Let us recall that the main difference between Algorithm 1 and Algorithm 2 lies in the sampling of the projections $\theta$ of the sliced Wasserstein. In the former, we resample $N_\theta$ projections at each iteration. Each sample is handled independently using the Gaussian mechanism. In the latter, we sample all $N_\theta$ projections

in advance and use them in all subsequent iterations. This approach allows for more iterations since the privacy budget is determined at initialization, but it requires a stricter privacy bound to ensure the same guarantees.

Indeed, the composition of DP operations used in Algorithm 1 enable the use of advanced composition techniques like the moment accountant (Abadi et al., 2016), resulting in tighter privacy bounds. In contrast, Algorithm 2 relies on basic composition, which is less efficient and yields looser bounds.

Let us give a mathematical intuition by expressing $\varepsilon_r$ and $\varepsilon$ respectively corresponding to the privacy budget for Algorithm 1 and Algorithm 2 for the same input noise $\sigma$. To ensure a fair comparison, we use the same number of projections in both cases. Given that, in Algorithm 1, we resample $T$ times the projections, we will be using $T * N_\theta$ as the number of projections for Algorithm 2 (already accounted for when we resample). As is, we will show that $\varepsilon_r < \varepsilon$.

**DPSWflow (Algorithm 2)**: Let us remind that the privacy guarantee is determined by the sensitivity of the function $\Delta'$ (how much a single data point changes the output) and the amount of noise added (via the noise scale $\sigma$). The privacy loss, measured as $\varepsilon$ is inversely proportional to $\sigma$:

$$\varepsilon \propto \frac{\Delta}{\sigma} \tag{52}$$

Therefore, by plugging in the sensitivity of the sliced Wasserstein gradient flow $\Delta' \propto \sqrt{\frac{N'_\theta}{d} + \sqrt{\frac{2N'_\theta(d-1)}{d+2}}}$ with $N'_\theta = T * N_\theta$, in the case of Algorithm 2 (DPSWflow), we would obtain the following:

$$\varepsilon^2 \propto \frac{TN_\theta}{\sigma^2 d} + \frac{1}{\sigma^2}\sqrt{\frac{2TN_\theta(d-1)}{d+2}} \tag{53}$$

**DPSWflow-r (Algorithm 1)**: However, the structure of this algorithm enables to use the moment accountant technique which gives a tighter bound for the privacy budget:

$$\varepsilon_r \propto \frac{q\sqrt{T}\Delta}{\sigma} \tag{54}$$

where $q = b/d$, $b$ being the batch size, $d$ the size of the dataset, $T$ is the number of iterations, and $\Delta \propto \sqrt{\frac{N_\theta}{d} + \sqrt{\frac{2N_\theta(d-1)}{d+2}}}$. We obtain the following:

$$\varepsilon_r^2 \propto \frac{q^2 TN_\theta}{\sigma^2 d} + \frac{q^2 T}{\sigma^2}\sqrt{\frac{2N_\theta(d-1)}{d+2}} \tag{55}$$

**Comparison**: Let us compare the privacy budget. $q = b/d$ is usually small: for relevant experiments, the dataset size tends to be large and batch size small. Comparing term by term we will have, for a reasonable number of iterations $T$:

$$q^2 TN_\theta < TN_\theta \quad \text{and} \quad q^2 T < \sqrt{T} \tag{56}$$

Therefore, we obtain $\varepsilon_r < \varepsilon$, which we also observe/compute in our experiments.

### C.5 Additional Comments on Hyperparameters and Algorithms

In this subsection we give all of the hyperparameters necessary for reproducing the experiments conducted.

**MNIST and FashionMNIST.** All three DPSWflow, DPSWflow-r, and DPSWgen models are evaluated with a batch size of 250 and for $\delta = 10^{-5}$ and the latent space (of the autoencoder, used as input of the mechanisms) has size 8. DPSWflow is evaluated over 1500 epochs for all values of $\varepsilon$, while DPSWflow-r

and DPSWgen are evaluated on 35 epochs for $\varepsilon = \infty$ and $\varepsilon = 10$, and on 20 epochs for $\varepsilon = 5$. DPSWflow uses $N_\theta = 31$ and $M_\theta = 25$, while DPSWflow-r and DPSWgen use $N_\theta = 70$. As explained in Section 4.3, the privacy tracker is different for DPSWflow compared to DPSWflow-r and DPSWgen. For DPSWflow we directly input the desired value for $\varepsilon$ and use the sensitivity bound from Eq. (23), along with $\sigma \geq c\frac{\Delta_2(f)}{\varepsilon}$ (from Section 2.3), to obtain the value of $\sigma$ which is used in our code. For DPSWflow-r and DPSWgen we used the following pairs of $\sigma, \varepsilon$: $\sigma = 0, \varepsilon = \infty$; $\sigma = 0.67, \varepsilon = 10$; and $\sigma = 0.8, \varepsilon = 5$.

**CelebA.** All three DPSWflow, DPSWflow-r, and DPSWgen models are evaluated with a batch size of 250, for $\delta = 10^{-6}$ and the latent space (of the autoencoder, used as input of the mechanisms) has size 48. DPSWflow is evaluated over 2000 epochs for every value of $\varepsilon$, while DPSWflow-r and DPSWgen are evaluated on 30 epochs for every $\varepsilon$. DPSWflow uses $N_\theta = 250$ and $M_\theta = 220$, while DPSWflow-r and DPSWgen use $N_\theta = 300$. For DPSWflow-r and DPSWgen we used the following pairs of $\sigma, \varepsilon$: $\sigma = 0, \varepsilon = \infty$; $\sigma = 0.58, \varepsilon = 10$; and $\sigma = 0.72, \varepsilon = 5$.

Also, notice that the architecture of the generator of DPSWgen is structured as follows: the input layer consists of a linear layer that transforms the input vector to a 256-dimensional vector. The first hidden layer applies a ReLU activation function to introduce non-linearity. The second hidden layer includes another linear layer that increases the dimensionality from 256 to 512, followed by a Batch Normalization layer to stabilize and accelerate the training process, and a ReLU activation function. The third hidden layer contains a linear layer that reduces the dimensionality from 512 to 256, followed by a ReLU activation function. The output layer comprises a final linear layer that maps the 256-dimensional vector to the desired output vector. This sequential arrangement of layers allows for progressive transformation and refinement of the input data, leveraging non-linear activation functions and normalization techniques to improve the model's learning capacity and stability.

