# OpenReview forum: "Differentially Private Gradient Flow based on the Sliced Wasserstein Distance"
_TMLR — Accepted by TMLR_

### Review · Reviewer_q2Vr · 2024-09-03

**Summary Of Contributions:**

The paper introduces a novel differentially private generative modeling approach based on gradient flow. The main contributions are the following:
- The authors define the gradient flow of the Gaussian-smoothed Wasserstein Distance and the associated SDE
- The authors show a link between smoothing and differential privacy based on a Gaussian mechanism
- In particular, the authors present two algorithms that implement their gradient flow
- The also analyze the differential privacy guarantee of their proposed particle scheme
- They perform experiments to show/validate their results.

**Audience:**

Yes

**Claims And Evidence:**

Yes

**Requested Changes:**

N/A

**Strengths And Weaknesses:**

Strengths:
- The paper is written very clearly and from the beginning, it is clear what the contributions of this paper are.
- The theoretical results are strong and the contribution is novel, as the authors say it has not been done before
- The experiments clearly highlight that their method is practical and viable.

---

> ### Author Response · Authors · 2024-11-28
> **Answer to reviewer q2Vr**
>
> We would like to thank the reviewer for their positive review. We appreciate the reviewer’s clear summary of our work. Their understanding accurately reflects the key aspects of our contributions, particularly the novel definition of the gradient flow for the Gaussian-smoothed Wasserstein Distance, our privacy assessment of the different smoothing operations, and the development and analysis of our algorithms. We are grateful for the reviewer’s positive comments on the clarity of the writing, the strength and novelty of the theoretical contributions, and the practical demonstration through our experiments.

---

### Review · Reviewer_YyB2 · 2024-11-05

**Summary Of Contributions:**

The paper studies DP generative modeling in the space of probability measures. They define the GF of the Gaussian-smoothed sliced Wasserstein distance and the natural stochastic process involved with it. They use natural discretization based on Euler's method to solve the continuous-time SDE. The DP guarantee of the corresponding GF is more straightforward after this.

**Audience:**

Yes

**Broader Impact Concerns:**

The paper has some interesting new directions, and I believe it definitely deserves acceptance.

**Claims And Evidence:**

Yes

**Requested Changes:**

Please see above and the pointed questions.

**Strengths And Weaknesses:**

The authors study an interesting problem and their solution is also very interesting. In particular, considering the minimization problem on the space of probability measures, this is less understood compared to DP-SGD. I think that the theoretical guarantees proved in this paper might lead to better performance than the generator-based model.

The proof in the paper also has some novel attributes. For example, they show the existence and regularity of solutions to the functional minimization problem which I am not sure exists in the literature.

I vote for acceptance. I however have a few clarifying questions.

I am trying to understand the proof of the regularity of the minimizer. Can the authors give more details as its relation to previous works or is the proof entirely new? If it is completely new, I feel that the authors undersell the importance of this result.

Define strong solution to the SDE before stating Theorem 2. If it was defined earlier, I have missed it. Mention somewhere that the assumptions of strong solution is standard in SDE literature.

Why do you get a tighter bound for Algorithm 1? Is there an intuitive reason for it?

---

> ### Author Response · Authors · 2024-11-28
> **Answer to Reviewer YyB2 1/2**
>
> We would like to thank the reviewer for their positive review. We address their questions below.
>
> ### Proof of the regularity
> > Give more details as its relation to previous works or is the proof entirely new? If it is completely new, I feel that the authors undersell the importance of this result.
>
> We clarify that Theorem 1 analyzes the first variation of the squared Gaussian-smoothed SW metric Eq. (12). To this extend, one first must establish the existence and regularity of solutions to the functional minimization problem Eq. (10). For this, we could not directly apply previous work.
>
> Indeed, the GSW metric is **not a simple application of the SW metric to Gaussian convoluted measures** (as mentionned in Appendix A, page 16). The convolution with Gaussian measure in the GSW metric occurs within the surface integral, separately for each one-dimensional projection of the original measures. To reiterate, $\mathcal{G}\_\sigma \mathcal{SW}^2\_2(\mu, \nu)  \neq \mathcal{SW}^2\_2(\mu*\xi_\sigma, \nu*\xi\_\sigma)$, where $\xi_\sigma \sim N(0,\sigma)$. We highlighted this in the revised version of the paper.
>
> Due to the distinct nature of the GSW metric, proving the existence and regularity of the minimizers to the functional in Eq. (10) has never been done before and **constitutes a novel contribution**. Proving both the regularity and existence of the minimizer could not be directly obtained from the DP-less gradient flow proof of *Liutkus et al. (2019)*. To establish regularity (Lemma 3), we adopt proof strategies inspired by *Bonnotte (2013)*’s previous work, but we need to adjust them to fit our specific GSW case. While the approach is informed by prior methods, **the proof requires a complete construction from scratch**, as it is not a straightforward adaptation.
>
>
> • *Liutkus et. al. Sliced-Wasserstein flows: Nonparametric generative modeling via optimal transport and diffusions. ICML 2019.*
> • *Nicolas Bonnotte. Unidimensional and evolution methods for optimal transportation. 2023*
>
>
> ### Strong solution definition
> > Define strong solution to the SDE before stating Theorem 2.
>
> A strong solution to an SDE means that it is defined over a *given* probability space and Wiener process, while a weak solution would be defined over *some* probability space and Wiener process. The assumption of strong solution is standard, as seen e.g. in *Liutkus et al. (2019)*. We included a relevant reference on SDEs for the interested reader *(Øksendal, 2000)*.
>
> • *Liutkus et. al. Sliced-Wasserstein flows: Nonparametric generative modeling via optimal transport and diffusions. ICML 2019.*\
> • *Øksendal. Stochastic Differential Equations: An Introduction with Applications. 2000*

---

> > ### Author Response · Authors · 2024-11-28
> > **Answer to Reviewer YyB2 2/2**
> >
> > ### Impact of resampling the projections on the privacy budget
> >
> > > Why do you get a tighter bound for Algorithm 1? Is there an intuitive reason for it?
> >
> > We thank the reviewer for this interesting question. We provide an answer below that we included in our revision.
> >
> > Let us recall that the main difference between Algorithm 1 (DPSWflow-r) and Algorithm 2 (DPSWflow) lies in the sampling of the projections $\theta$ of the sliced Wasserstein. In the former (DPSWflow-r), we resample $N\_\theta$ projections at each iteration. Each sample is handled independently using the Gaussian mechanism.  In the latter (DPSWflow), we sample all $N\_\theta$ projections in advance and use them in all subsequent iterations which introduces potential dependencies in the projections at different iterations. This enables us to use more iterations as the privacy budget is computed at initialization but the dependency increases the mechanism's sensitivity to data changes, needing a stricter privacy bound to maintain the same guarantees.
> >
> > Indeed, as explained in Section 3.3.3, the composition of DP operations used in Algorithm 1 is cheaper in privacy than doing them all at once at the beginning of the gradient flow as in Algorithm 2. Independent projections mean the privacy cost $(\varepsilon, \delta)$ can be composed using advanced composition techniques like the moment accountant (*Abadi et al. 2016*), resulting in a tighter bound, while correlated projections over iterations necessitate using basic composition, which is looser.
> >
> > Let us give a mathematical intuition. To this extent we will express $\varepsilon_r$ and $\varepsilon$ respectively corresponding to the privacy budget for Algorithm 1 (DPSWflow-r) and Algorithm 2 (DPSWflow) for the same input noise $\sigma$. To ensure a fair comparison, we use the same number of projections in both cases. Given that, in DPSWflow-r, we resample $T$ times the projections, we will be using $T*N_\theta$ as the number of projections for DPSWflow (it is already accounted for in DPSflow-r). As is, we will show that $\varepsilon_r < \varepsilon$.
> >
> > #### Algorithm 2
> >
> > Let us remind that the privacy guarantee is determined by the sensitivity of the function $\Delta'$ (how much a single data point changes the output) and the amount of noise added (via the noise scale $\sigma$).  The privacy loss, measured as $\varepsilon$ is inversely proportional to  $\sigma$: $$\varepsilon \propto \frac{\Delta}{\sigma}.$$
> >
> > Therefore, by plugging in the sensitivity of the sliced Wasserstein gradient flow
> > $\Delta' \propto  \sqrt{ \frac{N'\_\theta}{d} + \sqrt{   \frac{2N'\_\theta (d-1)}{d+2}}}$ with $N'\_\theta = T*N\_\theta$, in the case of Algorithm 2, we would obtain the following: $$\varepsilon^2 \propto \frac{TN\_\theta}{\sigma^2d} + \frac{1}{\sigma^2} \sqrt{   \frac{2TN\_\theta (d-1)}{d+2}}.$$
> >
> >
> > #### Algorithm 1
> >
> > However, the structure of the resampling in Algorithm 1 (DPSWflow-r) enables to use the moment accountant technique which gives a thighter bound for the privacy budget: $$ \varepsilon_r \propto \frac{q\sqrt{T}\Delta}{\sigma},$$ where $q=b/d$, $b$ being the batch size, $d$ the size of the dataset, $T$ is the number of iterations, and $\Delta \propto  \sqrt{ \frac{N_\theta}{d} + \sqrt{   \frac{2N_\theta (d-1)}{d+2}}}$. We obtain the following: $$\varepsilon^2_r \propto \frac{q^2 T N_\theta}{\sigma^2 d} + \frac{q^2 T}{\sigma^2}\sqrt{   \frac{2N_\theta (d-1)}{d+2}} $$
> >
> > #### Comparison
> >
> > Let us compare the privacy budget. $q=b/d$ is usually small: for relevant experiments, the dataset size tends to be large and batch size small. Comparing term by term we will have, for a reasonable number of iterations $T$:
> >
> > $$q^2TN_\theta < TN_\theta \text{  and  } q^2T < \sqrt{T}$$
> >
> > Therefore, we obtain $\varepsilon_r < \varepsilon$, which we also observe/compute in our experiments.
> >
> > • *Abadi et al. Deep Learning with Differential Privacy. ACM 2016.*

---

### Review · Reviewer_o1Ec · 2024-11-20

**Summary Of Contributions:**

This paper explores an alternative approach for training a DP generator other than DP-SGD, i.e. a gradient flow of sliced Wasserstain Distance (SWD). Particularly, a smoothed version of SWD is automatically a Gaussian mechanism with DP guarantees, so we can use it to train a DP generator. A formal DP guarantee is analyzed for each update. Experiments are conducted on both toy dataset and real image datasets.

**Audience:**

Yes

**Claims And Evidence:**

Yes

**Requested Changes:**

Please see above

**Strengths And Weaknesses:**

Self-claim: I know DP generator very well, but I do not know too much about the gradient flow/SDE part (so please refer to other reviewers for this part).

## Strength
1. Training DP generator via gradient flow is new, as far as I know.
2. This paper includes a complete theoretical analysis and experiments on toy and real datasets. Algorithm1, figures, and tables are clear to follow.

## Weakness
1. Motivation:
    + I do not find sufficient justification for choosing gradient flow as the generative modeling method, other than "it is under-explored". What is the benefit of this method?
    + Is there any other motivation for choosing Gaussian-smoothed SWD (I mean other than DP)?
3. Writing of gradient flow/SDE: Probably because I have limited knowledge in this field, I still do not get the connection between gradient flow and SDE. For example,
    + "Depending on the form of $F_\lambda$, this continuity can be associated with an SDE...". I cannot get how they are associated.
    + What is the connection between eq. 11, 13 and 17. Is eq. 17 an existing result? If so you need to add reference.
4. Discussion with diffusion models: diffusion models are also strongly connected with SDE, e.g. reversing the backward diffusion process is to find a solution to an SDE. What is the difference between gradient flow and diffusion?
2. Baselines: the experimental comparison is not complete. The only baseline is DPSWD, which I initially thought was similar to the proposed method. However, they are "fundamentally distinct". If so, why other distinct methods are not compared? For example, you can find a lot more related works in the numerical comparison table in [Dockhorn et al, 2023].

---

> ### Author Response · Authors · 2024-11-28
> **Answer to Reviewer o1Ec 1/3**
>
> We would like to thank the reviewer for their constructive review. We address their concerns below.
>
> ### 1. Motivation
>
> #### Gradient flow
>
> > I do not find sufficient justification for choosing gradient flow as the generative modeling method, other than "it is under-explored". What is the benefit of this method?
>
> In differential privacy for generative modeling, the predominant technique is using DP-SGD. The only alternatives involve generator-based methods that use a DP metric to train the generator. We propose a better approach using gradient flows.
>
> Wasserstein gradient flows present a natural framework for optimizing over the space of probablitity distributions. They are a viable alternative to other generative models in the non-DP setting *(Fan et al., 2022)*. Their mathematical framework excels in modeling the dynamics of distributions over time and inherently incorporate principled optimization techniques. These properties make them particularly well-suited for capturing complex data distributions while maintaining stability during training. For instance:
>
> - Gradient flows provide a continuous-time approximation of discrete methods like SGD and provides a baseline for designing and analyzing algorithms. In our case, it enables the design of a multi-step push-forward operator that transports a noisy distribution to the target distribution, rather than relying on a one-step push-forward function. This approach also offers valuable insights into the effects of discretization.
> - Optimizing over the space of probability distribution is a difficult problem and the regulatization term introduced by the flow helps in smoothing the probability path towards the target distribution.
> - As discussed below, gradient flows minimize a known metric, which makes integrating differential privacy straightforward, unlike diffusion models, where privacy must often be enforced using DP-SGD.
>
> Hence, leveraging them in a privacy-focused framework seemed like a natural progression. We argue in the introduction then confirm later in the paper that they should be better, privacy-wise, than generator methods. Indeed, we observe that for the same $\varepsilon$ and for the same exposure to data*, our DP gradient flow achieves better performance than a generator-based model trained with the same private metric. We suspect this is because the gradient flow brings a natural regularization leading to a more regularized optimization problem compared to generator learning.
>
> We will add these details in the revised version of the paper.
>
> (*)Meaning that $\varepsilon$ is obtain the same way (same number of epochs, same number of projections, same batch size: they observe the same data at the same moment.)
>
> • *Fan et al. Variational Wasserstein gradient flow. ICML 2022.*
>
>
>
>
> #### Metric
>
> > Is there any other motivation for choosing Gaussian-smoothed SWD (I mean other than DP)?
>
> Gaussian-smoothed Sliced Wasserstein Distance (SWD) is particularly well-suited for gradient flows compared to metrics like Sinkhorn or Wasserstein due to its computational efficiency and scalability, especially in high-dimensional settings *(Peyré & Cuturi 2019, Bonneel et. al. 2014, Bonet. 2022)*. By projecting distributions onto 1D slices, it avoids the curse of dimensionality while providing smoother and more stable gradients.  The Gaussian smoothing further enhances robustness to noise and outliers, ensuring well-behaved optimization dynamics *(Rabin et. al. 2012, Kolouri et al. 2019)*. Unlike Sinkhorn, which alters the true Wasserstein geometry with entropy regularization, SWD preserves the essential structure, making it an efficient and reliable choice for generative modeling tasks.
>
> • *Rabin et. al. Wasserstein Barycenter and Its Application to Texture Mixing. 2012.*\
> • *Bonneel et. al. Sliced and Radon Wasserstein Barycenters of Measures. Journal of Mathematical Imaging and Vision 2014*\
> • *Peyré & Cuturi. Computational Optimal Transport: With Applications to Data Science. Foundations and Trends in Machine Learning 2019.* \
> • *Kolouri et al. Generalized Sliced Wasserstein Distances, NeurIPS 2019.*\
> • *Bonet. Efficient Gradient Flows in Sliced-Wasserstein Space. TMLR 2022.*

---

> > ### Author Response · Authors · 2024-11-28
> > **Answer to Reviewer o1Ec 2/3**
> >
> > ### 2. Gradient flows & SDEs
> >
> > > "Depending on the form of $F\_\lambda$, this continuity can be associated with an SDE...". I cannot get how they are associated.
> >
> >
> > Let us go through the relation between $\mathcal{F}\_\lambda$ and the associated SDE. Recall that gradient flows aim at minimizing a functional $\mathcal{F}\_{\lambda}$ with:
> >
> > $$ \mathcal{F}_{\lambda} = \mathcal{F} + \lambda \mathcal{H},$$
> >
> > where $\mathcal{F}$ is the functional to be minimized and $\mathcal{H}$ the entropic regularization (with strength $\lambda \in \mathbb{R}$).
> >
> > A Wasserstein gradient flow constitutes a continuous sequence $(\mu_t)\_t$ of probability distribution within a Wasserstein metric space. $\mathcal{F}\_{\lambda}$ decreases along $(\mu_t)\_t$, more formally: the sequence follows a continuity equation (*Santambrogio, 2016*) with a general form of:
> > $$\frac{\partial\rho\_t}{\partial t} = \mathrm{div}(\rho\_t \nabla\_{W_2}\mathcal{F}\_{\lambda}(\rho_t) ) = \mathrm{div}(\rho_t \nabla\_{W_2}\mathcal{F}(\rho_t) ) + \lambda \Delta {\rho_t}  \:\:\:\:\:\:\:\: (5)$$
> >
> > where $\rho_t$ is the density of the probability flow $(\mu_t)\_{t\geq 0}$ at time $t$ and $\nabla\_{W_2} \mathcal{F}(\rho_t)\colon \mathbb{R}^d \to \mathbb{R}^d$ is the Wasserstein gradient of $\mathcal{F}$.  Or, one can notice that the above formulation for $\frac{\partial\rho_t}{\partial_t}$ **is a Fokker Planck type equation**. From there, we can obtain an SDE Eq. (6) on particles $(X_t)\_{t\geq0}$ whose densities provably follow Eq. (5), i.e. $X_t \sim \rho_t$  *(Jordan et al., 1998)*:
> > $$dX\_t = - \nabla\_{W_2}\mathcal{F}(\rho_t)(X_t)dt + \sqrt{2\lambda}dW_t   \:\:\:\:\:\:\:\: (6)$$
> > In our paper, we interchangeably used $\nabla V = \nabla_{W_2}\mathcal{F}(\rho_t)$. In our revision, we removed any mention of $V$ to clarify our explanation. We apologize for the confusion.
> >
> > Informally, this existing result means that if we can simulate the above SDE, then the distribution of $X_t$ would converge to the solution of the minimization of the functional $\mathcal{F}_\lambda$. There, $(X_t)_t$  could be used as samples drawn from $(\mu_t)_t$.
> >
> >
> > • *Santambrogio Euclidean, Metric, and Wasserstein gradient flows: an overview. 2016*\
> > • *Jordan et al.  The variational formulation of the Fokker-Planck
> > equation. SIAM Journal on Mathematical Analysis 1998*
> >
> >
> > > What is the connection between eq. 11, 13 and 17. Is eq. 17 an existing result? If so you need to add reference.
> >
> > The reviewer enquires about how Eq. (17) can be obtained from Theorem 1 (Eq. (11, 12, 13)). It consists in applying the SDE equivalence explained above, as we describe below.
> >
> >
> > Recall that, in our work, we aim at minimizing the functional described in Eq (10). in the paper:
> >
> > $$\mathcal{F}\_{\lambda, \sigma}^\nu(\mu) = \frac{1}{2} \mathcal{G}\_\sigma \mathcal{SW}\_2^2 (\mu, \nu) + \lambda \mathcal{H}(\mu)\:\:\:\:\:\:\:\: (10)$$
> >
> > With Theorem 1, we find the Wasserstein gradient flow and the associated continuity equation for Eq. (10) in Eq. (11):
> >
> > $$\frac{\partial\rho_t}{\partial_t} = - \mathrm{div}(v_t^{(\sigma)} \rho_t) + \lambda \Delta{\rho_t}, \:\:\:\:\:\:\:\: (11)$$
> >
> > $v_t$ thus corresponds to minus the Wasserstein gradient of the Gaussian smoothed sliced Wasserstein and is defined through Eq. (12) and Eq. (13).
> >
> > Then, using the above results and noticing that Eq. (11) is a Fokker Planck type equation: we can find an SDE on particles with corresponding densities. Thus, we consider a stochastic process $(X_t)_t$ that is the solution of the SDE starting at $X_0 \sim \mu_0$: $dX_t = v(X_t, \mu_t) dt + \sqrt{2 \lambda }dW_t$. It means that if we simulate the above SDE, then the distribution of $X_t$ would converge to the solution of the minimization of the functional of Eq. (10).
> >
> > In practice, the SDE $dX_t = v(X_t, \mu_t) d_t + \sqrt{2 \lambda }dW_t$ is a continuous time process, so it needs discretization. To this extend, we use a particle flow system that can be written as a collection of $N$ SDEs:
> >
> > $$dX_t^i = v(X_t^i, \mu_t^N) d_t + \sqrt{2 \lambda }dW_t^i, i=1,...N\in \mathbb{N}_+$$
> >
> > where, $\mu_t^N = \frac{1}{N} \sum_{j=1}^N \delta_{X_t^j}$ denotes the empirical distribution of the particles $\{X_t^j\}^N_{j=1}$.  Each particle of the above system can be simulated with Euler Marumaya discretization scheme, ultimately leading to Eq. (17) in the paper:
> > $$ \hat{X}\_{k+1} = h{v}^{(\sigma)}(\hat{X}\_k, \widehat{\mu}\_{kh})  + \sqrt{2\lambda h} Z\_{k+1}  \:\:\:\:\:\:\:\: (17)$$
> >
> >
> > **Summary of the connection between eq. 11, 13 and 17:** \
> > (11) : Continuity equation defined by the Wasserstein flow of the Gaussian Smoothed (depends on the derivative of the Kantorovich potentials).\
> > (13): Expression for the derivative of the Kantorovich potentials which a clear connection to the $v_t$ and the Gaussian smoothing. \
> > (17): Single SDE of the a particle scheme used to solves the minimization of the functional. The ensemble of these SDE constitues our generative model.

---

> > > ### Author Response · Authors · 2024-11-28
> > > **Answer to Reviewer o1Ec 3/3**
> > >
> > > ### 3. Diffusion models vs gradient flows
> > > > Diffusion models are also strongly connected with SDE, e.g. reversing the backward diffusion process is to find a solution to an SDE. What is the difference between gradient flow and diffusion?
> > >
> > > Gradient flows and diffusion models, while both relying on SDEs, differ fundamentally in their mathematical formulation and objectives.
> > >
> > > Gradient flows describe the steepest descent of a functional, often in a metric space like the Wasserstein space, and directly minimize an energy functional (e.g., the Wasserstein distance). Depending on the functional considered, a gradient flow can be related to an SDE through its Fokker-Planck formulation. In contrast, diffusion models rely on SDEs to describe transitions in data distributions, where the backward process explicitly reverses a forward diffusion process.
> > >
> > > Key differences lie in the nature of these SDEs.
> > > - While diffusion models have pre-defined probabilistic paths constituted with the data distribution noised at various levels, the same is not true for gradient flows which can admit more complex probability paths, albeit harder to eludidate.
> > > - Gradient flows converge asymptotically, as they perform a steepest descent, whereas the generating SDE of diffusion models terminates in finite time.
> > > - Gradient flows minimize a known metric, which makes integrating differential privacy straightforward, unlike diffusion models, where privacy must often be enforced using DP-SGD.
> > >
> > > For a detailed discussion of the similarities and differences between gradient flows and diffusion models, we refer to the work of *Franceschi et al. (2023)*.
> > >
> > > • *Franceschi et al. Unifying GANs and score-based diffusion as generative particle models. NeurIPS 2023*
> > >
> > > ### 4. Baselines
> > > > The experimental comparison is not complete. The only baseline is DPSWD [DPSWgen], which I initially thought was similar to the proposed method. However, they are "fundamentally distinct". If so, why other distinct methods are not compared? For example, you can find a lot more related works in the numerical comparison table in [Dockhorn et al, 2023].
> > >
> > > We would like to clarify the distinctions between our work, DPSWgen, and the diffusion models proposed by Dockhorn et al. (2023), as well as the scope of our experimental comparisons.
> > >
> > > Our work presents a differentially private (DP) gradient flow, which fundamentally differs from DPSWgen and diffusion models in its privacy mechanism and generative process.
> > > - DPSWgen, our chosen baseline, is a generator-based model trained using a DP metric. While both DPSWgen and our method use the same metric to measure distributional similarity, they differ in how samples are generated: DPSWgen employs a one-step generative model that maps Gaussian noise directly to the target distribution. In contrast, our method generates samples through a gradient flow, iteratively transitioning from a Gaussian distribution to the target distribution, acting as a multi-step generator.
> > > - Dockhorn et al. (2023), on the other hand, propose diffusion models that achieve differential privacy using DP-SGD. This approach is fundamentally distinct from ours, as it relies on stochastic processes and backward diffusion mechanisms, whereas our method directly integrates privacy into the gradient flow framework.
> > >
> > > Regarding baselines, our study focuses on a specific claim: in a privacy setting, a gradient flow performs better than generator-based models trained with the same metric. This is why DPSWgen was chosen as the baseline, as it provides the most relevant comparison for validating our contribution.
> > >
> > > We will revise the manuscript to clearly articulate this focus and the rationale for our experimental design, emphasizing that our claim is specific to outperforming generator-based models using the same metric.

---

### Review · Reviewer_yUc6 · 2024-11-28

**Summary Of Contributions:**

The paper proposes an alternative generative modeling approach based on gradient flow, which is defined by Gaussian-smoothed Sliced Wasserstein Distance, including the associated SDE. The authors define a numerical scheme for solving this SDE, and analyze the differential privacy guarantee of the proposed gradient flow. Experiments are conducted the demonstrate the results.

**Audience:**

Yes

**Claims And Evidence:**

Yes

**Requested Changes:**

Please see above

**Strengths And Weaknesses:**

Strengths
- The paper propose an alternative DP generative modeling via gradient flow
- Theoretical analysis and empirical results are provided
- The paper is clear and easy to follow

Weaknesses
- What are the practical values of L, c1, c2 in Theorem 2? If these values are large, the penalty parameter \lambda will also be very large, making it impractical.
- The parameter \delta is used with different meanings in Sections 3.2 and 3.3.
- What is the data dimension d used in the experiments? While an autoencoder can reduce d for the provided image datasets, it may negatively impact performance on other types of data. It would be helpful to evaluate and illustrate performance with a larger d.

---

> ### Author Response · Authors · 2024-11-28
> **Answer to Reviewer yUc6**
>
> We would like to thank the reviewer for their constructive review and address their remarks below.
>
>
> ### 1. Constants in Theorem 2
>
> > What are the practical values of L, c1, c2 in Theorem 2? If these values are large, the penalty parameter $\lambda$ will also be very large, making it impractical.
>
> As is common in theoretical analyses of similar problems, the values of these constants are constrained by our assumptions and are not explicitly computable in practical settings. Their precise values depend on the specific properties of the system and the SDE being studied, which makes them challenging to determine in practice. Similarly to *Liutkus et al. (2019)* for their respective gradient flow, our primary goal with Theorem 2 is not to compute these constants. Instead, it is to provide a theoretical framework and the quality of the solution when discretizing the SDE compared to the continuous version, for a fixed $\lambda$. In this regard, our results align closely with those obtained by *Liutkus et. al. (2019)*, reinforcing the validity of our gradient flow and confirming that it behaves as expected within this theoretical context.
>
> Regarding the reviewer's concern on the value of $\lambda$ constrained by these constants, we recall that $T = Kh$ and $\lambda > TL^2/8$. Thus, for any value of $\lambda$, we can choose $h$ sufficiently small so that it satisfies the conditions.
>
>
>
> • *Liutkus et al. Sliced-Wasserstein flows: Nonparametric generative modeling via optimal transport and diffusions. ICML 2019.*
>
> ### 2. $\delta$'s Use
> > The parameter $\delta$ is used with different meanings in Sections 3.2 and 3.3.
>
>
> We apologize for the confusion caused by the inconsistent use of the parameter $\delta$. In the revised version of the paper, we have addressed this issue by renaming $\delta$ from Theorem 2 to $\kappa$. To clarify, in Theorem 2, $\kappa$ is a constant greater than $0$ and independent of time. In contrast, in the privacy definition, $\delta$ represents a bound on the external privacy risk. We hope this revision resolves the ambiguity.
>
>
> ### 3. Data Dimensions
> > What is the data dimension d used in the experiments? While an autoencoder can reduce d for the provided image datasets, it may negatively impact performance on other types of data. It would be helpful to evaluate and illustrate performance with a larger d.
>
> We understand the reviewer’s concern regarding the data dimension $d$ used in our experiments. The specific values of $d$ are given in Appendix C5 (page 24): $d=8$ for MNIST and FashionMNIST and $d=48$ for CelebA.
>
> It is true that performance may vary with other types of data, but our study deliberately concentrates on images because they constitute the standard benchmark modality in generative modeling. For images, the use of autoencoders for dimensionality reduction is well-established in the literature (e.g., in Stable Diffusion - *Rombach et. al. (2022)*).
>
> Finally, we acknowledge that the sliced Wasserstein distance is sensitive to the dimensionality $d$, and a smaller $d$ is necessary for stability in our approach. However, this does not affect the validity of our contributions or claims, which remain focused on the theoretical and methodological aspects -- comparing gradient flow and generator-based model for the same DP metric -- rather than the data type or dimensionality.
>
> • *Rombach et. al. High-Resolution Image Synthesis with Latent Diffusion Models. CVPR 2022.*

---

### Author Response · Authors · 2024-11-28
**Global Comment**

We would like to thank the reviewers for their constructive feedbacks. We appreciate that they recognized the **great interest and clarity** of our differentially private generative modeling approach. All reviewers endorse the clarity, theoretical correctness and novelty of our work, with Reviewer YyB2 appreciating the details on existence and regularity properties. Reviewers q2Vr and yUc6 also confirm the relevance of our empirical results.


We address the reviewers' comments in individual responses to their reviews below. Accordingly, we provided a **revision** of our submission with the following changes/additions, highlighted in red in the paper.
- **Reviewer YyB2**:
  - regularity proof - GSW metric (page 5 & Appendix A page 16),
  - strong solution (page 6),
  - intuition on why it is better to resample for DP (Appendix C4 page 23 & 24).
- **Reviewer o1Ec**:
  - improved motivation for the gradient flow (page 2),
  - clarified explanation of the link between the gradient flow and the SDE (page 3),
  - clarified claim for experimental section / scope (pages 9 & 10).
- **Reviewer yUc6**: replaced constant $\delta$ by $\kappa$ in Theorem 2's statement and proof to avoid confusion with the privacy parameter sharing the same notation (Section 3.2 & Appendix B).


We look forward to further discussing with the reviewers before they provide their final recommendations.

---

> ### Author Response · Authors · 2024-11-28
> **Minor improvements in the paper's experimental results**
>
> We would like to report to the reviewers a **correction and slight improvement of our experimental results**. We apologize for the confusion and thank the reviewers in advance for considering this new element.
>
> We noticed during the rebuttal phase an oversight in the code, detailed below, that slightly impacted our calculations of the privacy guarantee $\varepsilon$ using the moments accountant. This affected the DPSWflow-r and DPSWgen models. Fortunately, after correction, the **impact remains minimal and our conclusions unchanged**. The paper and the code have been updated to reflect these adjustments, including revised FID scores (Table 1) and generated images: Figures 2,3,4 (in red in the revised version of the paper).
>
>
> The current and previously obtained results are detailed below.
>
> **MNIST**
> > $\varepsilon = 10$
> >    - DPSWgen : FID = 124 (previously 128)
> >    - DSWflow-r: FID = 70 (previously 71)
>
> > $\varepsilon = 5$
> >    - DPSWgen : FID = 198 (previously 203)
> >    - DSWflow-r: FID = 114 (previously 117)
>
> **FashionMNIST**
> > $\varepsilon = 10$
> >    - DPSWgen : FID = 170 (previously 172)
> >    - DSWflow-r: FID = 88 (previously 88)
>
> > $\varepsilon = 5$
> >    - DPSWgen : FID = 199 (previously 205)
> >    - DSWflow-r: FID = 98 (previously 99)
>
> **CelebA**
> > $\varepsilon = 10$
> >    - DPSWgen : FID = 209 (previously 209)
> >    - DSWflow-r: FID = 132 (previously 134)
>
> > $\varepsilon = 5$
> >    - DPSWgen : FID = 214 (previously 215)
> >    - DSWflow-r: FID = 197 (previously 202)
>
>
> The mistake has been corrected in the code (`DPSWflow/flow/swf_moment.py`, line 137; `DPSWgen/model/train.py`, line 89) with by replacing:
>
>                 log_moment += compute_log_moment(q_batch, sigma, T, lmbd)
>
> that wrongly added log moments over lambdas, with:
>
>                 log_moment = compute_log_moment(q_batch, sigma, T, lmbd)

---

### Decision · Action_Editor_8rLs · 2025-01-04

**Recommendation:** Accept as is

**Comment:**

Overall, the reviewers are not super enthused by the technical novelty of the paper, but overall they all agree that the paper's claims are validated by strong evidence.

**Audience:**

Subset of the community interested in differential privacy

**Claims And Evidence:**

The paper's claims are technically correct and corroborated by evidence. The contributions are not super novel, but the paper definitely passes the evaluation criteria at TMLR.